# PIECEWISE LINEAR PARAMETRIZATION OF POLICIES: TOWARDS INTERPRETABLE DEEP REINFORCEMENT LEARNING

**Maxime Wabartha**[*]
McGill University, Mila

**Joelle Pineau**
McGill University, Mila, FAIR at Meta

## ABSTRACT

Learning inherently interpretable policies is a central challenge in the path to developing autonomous agents that humans can trust. Linear policies can justify their decisions while interacting in a dynamic environment, but their reduced expressivity prevents them from solving hard tasks. Instead, we argue for the use of piecewise-linear policies. We carefully study to what extent they can retain the interpretable properties of linear policies while reaching competitive performance with neural baselines. In particular, we propose the HyperCombinator (HC), a piecewise-linear neural architecture expressing a policy with a controllably small number of sub-policies. Each sub-policy is linear with respect to interpretable features, shedding light on the decision process of the agent without requiring an additional explanation model. We evaluate HC policies in control and navigation experiments, visualize the improved interpretability of the agent and highlight its trade-off with performance. Moreover, we validate that the restricted model class that the HyperCombinator belongs to is compatible with the algorithmic constraints of various reinforcement learning algorithms.

## 1 INTRODUCTION

Artificial intelligence (AI) enables practitioners to automate processes by replacing human decisions and predictions with computational ones. The value potential is high, since the rules are learnt from data: predictions can be made more accurate (Silver et al., 2017; Nam et al., 2019) or quicker (Le-Cun et al., 1989; Lindsey et al., 2018) than their human counterparts. To achieve these feats, models have rapidly grown in complexity. This trend has come at the cost of interpretability: recent models are opaque decision makers. As a consequence, they are untrustworthy in many situations, either because their predictions are unstable (Goodfellow et al., 2014; Lu et al., 2020), fail to generalize (Wabartha et al., 2021), the actual sub-policies differ from the expected one (Amodei et al., 2016; Ibarz et al., 2018), or because one cannot easily extract the logic behind their decisions.

Often overlooked, in our quest for ever improving performance, the transparency and explainability of AI models might soon become an indispensable component of their democratization, amid a growing oversight by regulatory bodies (European Union, 2016; United States of America, 2022; United Kingdom, 2023a;b). But interpretability is also a fascinating scientific topic in itself, giving us the opportunity to strengthen our understanding of our models and their effects. Therefore, it is important to systematically explore the methods that let us interpret machine learning models. Deep reinforcement learning (DRL) is a particularly interesting domain in which to study interpretability. In fact, a precise description of how policies interact with an environment can inform other algorithms designed to improve the fairness, accountability, or transparency of the deployed model (Doshi-Velez & Kim, 2017). Transparent policies could also help better diagnose the agent's interactions, which can be helpful if it appears to malfunction. Such *functionally interpretable* policies are especially of interest as they do not need a separate algorithm providing potentially approximate explanations for a policy's decisions (*post-hoc interpretability*) (Rudin, 2019)).

In this work, we examine the viability of piecewise-linear policies as a beneficial compromise between interpretability and performance in DRL. Piecewise-linear policies such as multi layer perceptrons (MLPs) with ReLU activations directly extend linear policies, which hold favorable inter-

---

[*]Work partially done as visiting researcher for FAIR at Meta. `maxime.wabartha@mail.mcgill.ca`

pretability properties (Molnar et al., 2021). However, the extremely large number of linear sub-policies usually learnt by MLPs (Hanin & Rolnick, 2019) hinders their interpretability, even if many of these linear sub-policies might in fact be superfluous for end performance Akrour et al. (2018). We focus on extending the analysis of piecewise-linear policies in the context of interpretability. To that end, we propose the HyperCombinator (HC), a neural architecture that parametrizes a piecewise-linear policy with a controllably small number of linear sub-policies. The HC agent selects a linear sub-policy at every interaction with the environment, an important property missing from previous works. The advantages of our piecewise-linear implementation of the policy are many: we can first guarantee that the agent, which can be thought of as a mixture of linear experts (Jacobs et al., 1991), interacts with the environment exclusively through one of the learnt sub-policies. Moreover, each sub-policy can be inspected individually. Finally, we can probe the policy at any given interaction to transparently analyze the computation leading to the decision.

Our contributions are the following: (1) we characterize properties that an interpretable piecewise-linear policy should ideally satisfy, (2) we present the HyperCombinator, an actor architecture that implements a piecewise-linear policy with a specifiable number of linear sub-policies and compatible with most RL algorithms, (3) we characterize the interpretability properties inherited by HC, (4) we extensively evaluate the model on control and navigation tasks and observe a sustained performance of the model despite its greatly reduced expressivity. In addition, (5) we leverage the two levels of interpretability of HC to propose two visualizations. The first one lists the different ways the policy can react to an input by cataloging *exhaustively* each linear sub-policy and its coefficients into a table. The second condenses the sequence of decisions into the sequence of sub-policies used, surfacing the temporal abstractions emerging from the task.

## 2 BACKGROUND

### 2.1 DEEP REINFORCEMENT LEARNING AND INTERPRETABLE POLICIES

In reinforcement learning, an agent learns to interact with its environment to maximize the accumulation of a learning signal, the reward (Sutton & Barto, 2018). The environment is modeled as a Markov Decision Process (MDP) defined as a tuple $(\mathcal{X}, \mathcal{A}, R, P, \gamma, \rho_0)$. At each timestep $t$, the agent in state $x_t \in \mathcal{X}$ takes the action $a_t \sim \pi(x_t) \in \mathcal{A}$. The agent then transitions to $x_{t+1} \sim P(\cdot|x_t, a_t)$ and receives reward $r_{t+1} = R(x_t, a_t, x_{t+1})$. We denote with $\rho_0$ the starting state distribution and represent a trajectory $\tau$ as $(x_0, a_0, x_1, \ldots, x_T)$, where $T$ is the last timestep of the trajectory if the environment is episodic, and $T = \infty$ else. The discounted sum of rewards is $G(\tau) = \sum_{t=0}^{T-1} \gamma^t r_{t+1}$, and we denote its expectation $J^\pi = \mathbb{E}_\tau [G(\tau)]$. Fundamental in reinforcement learning is the function $Q^\pi$, for which for a given state and action, $Q^\pi(x, a) = \mathbb{E}_\pi \left[ \sum_{t=0}^{T-1} \gamma^t R(x_t, a_t, x_{t+1}) | x_0 = x, a_0 = a \right]$. Actor-critic algorithms solve an MDP by modelling explicitly both $\pi$ and $Q^\pi$. They aim at finding a policy $\pi^*$ such that $\pi^* = \mathrm{argmax}_\pi J^\pi$.

Both $Q^\pi$ and $\pi$ can be parametrized by deep neural networks (NNs) to handle high-dimensional input spaces and increase the performance of the agent (Mnih et al., 2015). However, this modelisation limits our understanding of the action choice of an agent, due to the opacity of NNs.

In this work, we tackle the task of making the *policy* interpretable at each timestep. Since we focus on functional interpretability, we constrain the class of functions that the policy belongs to. That is, we design a policy architecture that gives exact insights about the decision applied to state $x_t$. The last layer of deep policies is often a non-linearity $\mu$, such as the softmax function, to cast the prediction of the NN to the right action space. In a similar fashion to the interpretation of logistic regression, we focus on interpreting the *pre* non-linearity part of the policy, which we denote by $\tilde{\pi}$. Finally, we remark that an interpretable model is of limited utility if the input features themselves are not interpretable. Therefore, we consider robotics environments with varying levels of difficulty and where the input to the policy is the proprioceptive state representation, each feature representing a physical variable describing the agent (Tassa et al., 2018).

### 2.2 FROM LINEAR TO PIECEWISE-LINEAR POLICIES

To achieve functional interpretability, one can reduce the size of the class that the policy belongs to (Rudin, 2019). Classes of functions that can be expressed through a small number of parame-

ters, such as linear functions, are often considered to be interpretable due to the reduced amount of information required to understand the function they realize (Molnar et al., 2021). We write linear policies as $\tilde{\pi}(x) = \langle \theta, x \rangle$, where $\theta$ designates the *linear coefficients*, the input $x$ is assumed interpretable, and $\langle A, b \rangle$ is the dot product between $A$ and $b$ if both are vectors, and the matrix-vector product if $A$ is a matrix instead. Linear policies have several desirable properties related to interpretability, such as being summarizable by their coefficients, or having each coefficient explicitly quantify the influence of a given feature and expressing the same valid explanation for all inputs. However, despite showing strong performance in certain benchmarks (Rajeswaran et al., 2017), their flexibility is generally not sufficient to handle more complex tasks (Mnih et al., 2013).

We therefore consider the larger class of piecewise-linear policies. They are a direct extension of linear policies, as they can be written as $\tilde{\pi} : \mathcal{X} \to \mathcal{A}$ with:

$$\tilde{\pi}(x) = \begin{cases} \langle \theta_{\omega_0}, x \rangle & \text{if } x \in \omega_0 \\ \langle \theta_{\omega_1}, x \rangle & \text{if } x \in \omega_1 \\ \dots & \\ \langle \theta_{\omega_{K-1}}, x \rangle & \text{if } x \in \omega_{K-1} \end{cases}, \tag{1}$$

where $K$ is the number of linear sub-functions, and where the partition of the input space $\Omega = \{\omega_0, \dots, \omega_{K-1}\}$, the set of linear coefficients $\theta = \{\theta_\omega, \omega \in \Omega\}$ and the function mapping an input to the corresponding subset $\omega$, $a : \mathcal{X} \to \Omega$ describe $\tilde{\pi}$. Therefore, $\tilde{\pi}$ is locally linear, and some interpretability properties of linear policies carry over. However, both the value of $K$ and the complexity of $\Omega$ and $a$ influence the interpretability of $\tilde{\pi}$.

MLPs using only linear layers (including linear, convolutional or layer normalization layers) and ReLU activations (Fukushima, 1975; Nair & Hinton, 2010) already offer a piecewise-linear parametrization of policies (Pascanu et al., 2014) learnable with gradient descent. The rightmost plot of Fig. 1 illustrates the induced partition, and Hanin & Rolnick (2019) analyse more precisely the nature of the partition. Growing the width and the depth of the network quickly increases the approximation power of MLPs by increasing the number of linear sub-functions that compose them (Pascanu et al., 2014) and the complexity of the partition, in turn penalizing the interpretability of $\tilde{\pi}$. Conversely, it is only possible to loosely control an upper bound on the number of linear sub-policies through the architecture of the MLP (Hanin & Rolnick, 2019). Moreover, a policy with a large number of linear sub-policies implies that each sub-function only applies to a few inputs, limiting the generalization of the explanations provided by the linear coefficients. For instance, we show in Appendix C that even a baseline RL agent provided with a small neural policy uses a unique linear sub-function around 950 times in a 1000 step-long trajectory, meaning that a different set of coefficients is applied at almost every step. Restricting the number of linear sub-functions while maintaining a high level of performance and interpretability is therefore a major challenge, that we explore next.

## 3 METHODS

As noted in Sec. 2.2, piecewise-linearity is not a sufficient requirement to guarantee that we obtain an interpretable policy. Therefore, we begin by formulating properties that a deployed, interpretable policy should ideally satisfy.

**(P1) Sub-policy transparency**: the input to output sub-policy computation is easy-to-understand.
**(P2) Assignation transparency**: the input to sub-policy choice computation is easy to understand.
**(P3) Separation**: the policy uses a single sub-policy at each interaction (training and evaluation).
**(P4) Counterfactual reasoning**: we can compare the sub-policy taken to the other possible ones.
**(P5) Generalization**: the sub-policies generalize to similar inputs.
**(P6) Causality**: we want to justify the causal influence of each feature *w.r.t.* the task success.

Given the subjective nature of interpretability, a single definition might not be sufficient (Doshi-Velez & Kim, 2017), and several desiderata relate to different notions of interpretability. To satisfy as many of these properties as possible, we design the Hypercombinator, a piecewise-linear neural architecture with a small, controllable number of linear sub-functions.

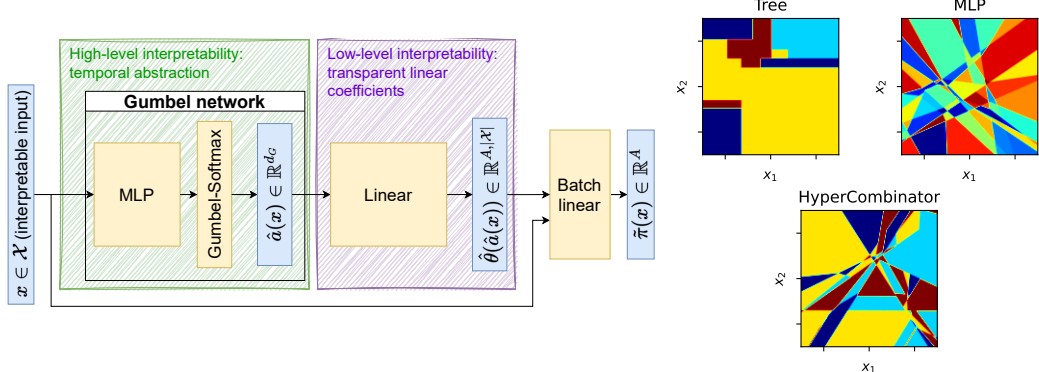

Figure 1: (Left) Proposed actor architecture. Layers are colored in yellow, vectors in blue. (Right) Intuition of the partition of a 2D space into linear regions by different models. Each polygon corresponds to a distinct linear region, and each color to a set of unique linear coefficients.

### 3.1 PROPOSED POLICY ARCHITECTURE

Our starting policy class is expressed by an MLP, realizing a piecewise-linear function with respect to an input $x \in \mathbb{R}^n$ assumed *interpretable*. Following Sec. 2.2, the prediction can be decomposed as $\tilde{\pi}(x) = \langle \theta_{a(x)}, x \rangle$, where $a : \mathbb{R}^n \to \Omega$ assigns an input $x$ to a subset of the partition induced by $\tilde{\pi}$. For a general MLP, both $a$ and $\theta$ are implicitly defined by $\tilde{\pi}$ and have a complex dependency on its parameters. This intricate relationship illustrates two major hurdles in making neural networks interpretable, namely the complexity of the induced partition and the plurality of linear sub-functions. Both relate to the complexity of the policy class.

**Parametrization** We focus on reducing the number of sub-policies while retaining the complexity of $\Omega$. To do so, we make the functions $a$ and $\theta$ explicit in the output computation through a new parametrization of the NN. We first view $\theta$ as a linear function $\hat{\theta} : \Omega \to \mathbb{R}^n$ rather than a set of coefficients. Secondly, we model $a$ explicitly within $\tilde{\pi}$ by an MLP followed by a Gumbel-Softmax layer (Jang et al., 2016; Maddison et al., 2016), and refer to the composition of both as a Gumbel network $\hat{a}$. The Gumbel-Softmax is a technique typically used to continuously learn a categorical variable. In this context, it predicts to which sub-policy should an input $x$ be mapped. Furthermore, we use the straight-through estimator (Bengio et al., 2013) to force $\hat{a}$ to produce strict assignations, *i.e.* one-hot encodings in the forward pass while being trainable with gradient descent. We note that because the output of the Gumbel network is always a one-hot encoding, we can use any activation function within the Gumbel network without breaking the piecewise-linear property of the policy. The resulting architecture is

$$\tilde{\pi}(x) = \langle \hat{\theta}(\hat{a}(x)), x \rangle, \tag{2}$$

$$\pi(x) = \mu(\tilde{\pi}(x)), \tag{3}$$

which we refer to as the HyperCombinator, illustrated in Fig. 1 along with the partition it induces.

This formulation is advantageous for several reasons. First, the MLP preceding the Gumbel-Softmax induces the partition of the input space which $\pi$ inherits, thus retaining some of the predictive flexibility of NNs. Second, we explicitly control the number of unique sub-policies of $\pi$ through the dimension $d_G$ of the Gumbel-Softmax layer. Moreover, the Gumbel network $\hat{a}$ is piecewise-constant thanks to the straight-through estimator. Therefore, $\pi$ interacts with the environment only through one of the $d_G$ sub-policies, on the contrary to previous work (Akrour et al., 2018).

Policies modeled with the HyperCombinator differ from MLPs in that they are usually not continuous at the border between linear regions, which we do not find to be a problem for stability in practice (Appendix D.6). Moreover, reducing the width or the height of the NN is an alternative and more direct way to reduce the number of unique coefficients. Yet, the number of linear sub-functions grows very quickly with the size of the architecture, both theoretically and empirically. We refer to Appendix B for a more in-depth study of this approach.

## 3.2 INTERPRETABILITY CHARACTERIZATION OF THE HYPERCOMBINATOR ARCHITECTURE

The HyperCombinator policy class has the following structural properties:

- $\tilde{\pi}$ is a linear function of the interpretable input $x$ (P1). A stakeholder can therefore access transparently the computation that led to the decision, given the sub-policy. The coefficients $\hat{\theta}(\hat{a}(x))$ are available as an output of our approach.
- The policy is piecewise-linear during evaluation *and* training thanks to the use of the straight-through estimator. This ensures that HC stays interpretable for all environment interactions (P3).
- The number of unique sub-policies of the policy is set by the hyperparameter $d_G$ (P4), which lets a stakeholder control the performance-interpretability trade-off.

Subsequently, our architecture needs to re-use the few sub-policies of very low complexity in order to solve complex tasks. The sub-policies are thus in practice constrained to generalize across a wide range of states, which implies P5. This is not true for MLPs, due to their huge number of linear regions and unique coefficients. MLPs additionally do not satisfy P4.

Our approach is *locally interpretable*, as it explains the sub-policy applied to any given example. Conversely, while the explanations generalize to the inputs that are mapped to the same sub-policy, the fact that the partition is induced by a neural network prevents us from having a *globally interpretable* model; our approach does not satisfy P2. As an important consequence, the HyperCombinator does not provide feature importance or any notion of causal explanations and does not satisfy P6. Indeed, just like MLPs, a feature $x_i$ of $x$ can be involved in the computation of all the coefficients $\hat{\theta}(\hat{a}(x))$ and simultaneously have the value of its linear coefficient $\hat{\theta}(\hat{a}(x))_i$ be 0. Instead, an HC policy provides a transparent justification of how it interacted with the environment for a given input, which is governed by the transparent sub-policy. In other words, our approach can be understood as *interpretable conditioned on the choice of a sub-policy*.

**Two levels of interpretability**  In summary, an HC policy can perhaps best be understood as two consecutive modules, each corresponding to a different level of interpretability. The first module selects a linear sub-policy given an input, while the second retrieves the linear coefficients of the selected sub-policy and applies them to the input. The second module provides the exact linear coefficients that are used by the policy to interact with the environment, quantifying the influence of each interpretable feature on the chosen action. We characterize this as the *low-level interpretability* of the HC policy. Simultaneously, one can observe the sequence of choices of sub-policies made by an agent in order to analyze its behavior, thanks to the first module of $\tilde{\pi}$. The forced re-use of sub-policies leads to the emergence of cycles and temporal abstractions which let us analyze how the agent solves the task, that we call *high-level interpretability*. We illustrate the complementary existence of both level of interpretabilities in Fig. 1 (left), using green and purple to respectively locate the high and low-level interpretability of $\tilde{\pi}$. We will see in Sec. 4 that these two levels of interpretability enable insightful visualizations of the agent's policy, such as the identification of the behavior modelled by each linear subpolicy (Appendix. D.7) and the observation of the emergence of temporal abstractions (Sec 4.1 and 4.2).

## 3.3 APPLICATION TO DEEP ACTOR-CRITIC ARCHITECTURES

We can use the HyperCombinator as the actor of any actor-critic algorithms, provided that no competing assumption is made on the architecture of the policy. This means that the critic or the potential other parts of the RL algorithm can be modeled by complex neural networks without affecting the interpretability of the policy. We note that in practice, all the linear sub-policies share the same bias vector. This limits the flexibility of the sub-policies, but also increases their interpretability since they all reduce to the same default sub-policy at $x = 0$. We adapt $\mu$ to the action space of interest. In addition, we note that the Gumbel-Softmax layer can destabilize training. For instance, its predictions may collapse to a single mode, making $\hat{\pi}$ equivalent to a linear policy. We alleviate this issue through different forms of regularization. First, we increase the temperature parameter and add weight decay to the last layer weights of the Gumbel network MLP. Both limit the magnitude of the input fed to the Gumbel-Softmax layer. Secondly, we maximize the entropy $H$ of the average sub-policy assignation over a batch of size $B$, that is $H(\frac{1}{B}\sum_{i=1}^{B}\hat{a}(x_i))$. This encourages the Gumbel network to use the diversity of sub-policies at its disposal.

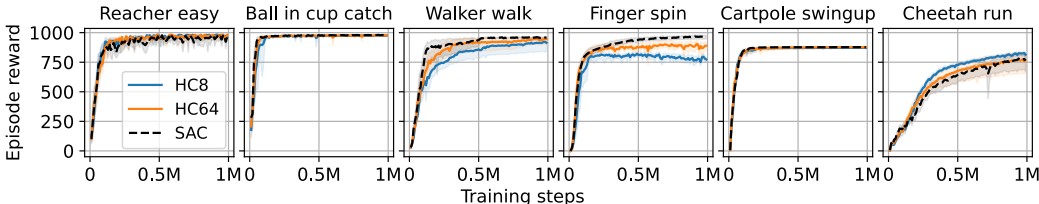

Figure 2: HC actors match or approach the performance of SAC for most environments.

## 4 EXPERIMENTS

In this section, we evaluate if the structural constraints of the HyperCombinator offer a reasonable compromise on performance, and visualize the improved interpretability properties of our approach.

### 4.1 CONTROL

We evaluate how well HC policies can control proprioceptive variables such as the joints of a robot through the DeepMind Control Suite benchmark (Tassa et al., 2018). We aim to (1) ensure that the HC architecture can be trained to solve control tasks, (2) investigate the improved interpretability of the agent and (3) compare the performance of HC actors to their baseline algorithm using their usual neural policy. We replace the actor in the SAC algorithm by a HC actor with final non-linearity $\mu$ which predicts, for state $x$, the mean of the action distribution $\pi(x) = \mu(\tilde{\pi}(x))$. In other terms, $\tilde{\pi}$ is linear *w.r.t.* the interpretable proprioceptive features (Sec. 2.1). The Gumbel network follows a $[1024, 1024, 1024]$ architecture and uses a sub-policy assignation entropy coefficient of $0.001$. The remaining hyperparameters are the defaults from Yarats & Kostrikov (2020), that we detail in Appendix D. We consider 2 variants of the HC actor, HC8 and HC64, with a maximum of respectively 8 and 64 sub-policies. The latter model is more expressive, at the cost of interpretability, since a practitioner now needs to factor in 64 different possible sub-policies instead of 8.

**Performance** We perform an in-depth experimental analysis of the performance of piecewise-linear policies in DM Control, extending previous work (Akrour et al., 2018). Figure 2 presents the evolution of the evaluation return over training for a subset of 6 commonly used environments (Yarats et al., 2021), while we defer the results for all environments to Appendix D.2. HC policies approach or match the performance of SAC in most environments, with Inter-Quartile Mean (Agarwal et al., 2021) scores (95% CI) of .81 (.78-.83), .85 (.83-.87) and .87 (.86-.88) respectively for HC8, HC64 and SAC. We highlight in particular the limited drop in performance of HC8, which performs at most 3% worse than SAC in 57% of the tested environments (HC64: 78%). These results show that drastically reducing the expressivity of the actor is only penalized by a moderate drop in performance: HC8 achieves a comparable return to SAC in most environments despite having restricted the interaction capability of the actor to only 8 different linear functions. We can improve further these results by trading some of the interpretability of the agent, as illustrated by the results of HC64. We further analyze the performance-interpretability gap in Appendix D. In addition, the HC architecture requires the sub-policies to be reused for a large number of inputs, which can help sample efficiency. We observe this effect in the Cheetah run and Humanoid tasks after training for 5M time-steps. We find that HC8 and HC64 improve over SAC during early parts of the training in 3 of these tasks. In the Humanoid Stand environment in particular, HC8 and HC64 converge several million steps before SAC. We further illustrate these results in Appendix D. Conversely, the performance curves of HC policies suffer a higher variance than SAC's, which we presume is due to the presence of the Gumbel-Softmax layer in the network.

**Interpretability** The low-level interpretability of HC actors, thanks to the linearity of the sub-policies, gives insights about the influence of the features in each decision. In the Cartpole swingup environment, we can for instance precisely analyze how some sub-policies swing the cartpole up while others stabilize the cartpole once in an approximately straight position. We defer the precise analysis of this result to Appendix D.7. The discrete bottleneck introduced by the Gumbel network also lets us visualize the high-level interpretability of HC through the sequence of chosen sub-

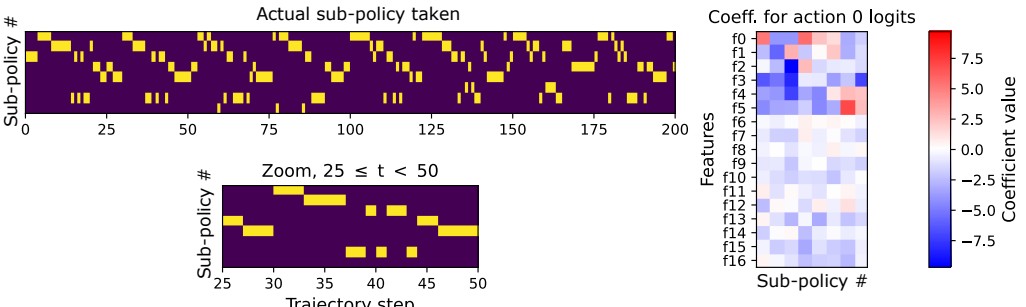

Figure 3: (Left) Sub-policy sequence of the first 200 timesteps of HC8 on Cheetah run. A temporal abstraction emerges as HC8 reuses the same sub-policy for several timesteps before switching to a new one in a cyclical fashion. (Right) All sub-policies coefficients, for one dimension of $\tilde{\pi}$'s output space. This view exposes the similarities of sub-policies (2 and 3 have same-sign coefficients for features 1 to 5) or their disparities (2 and 3 have opposite sign (hence effect) *w.r.t.* feature 0.

policies. For instance, HC8 learns to control the joints of a robot to maximize its speed in the Cheetah run environment. The learnt movement is cyclical by nature, which is adapted to the strong sub-policy re-use of HC. We find again this periodicity in the sequence of sub-policies taken by HC8, in the left part of Fig. 3. The successive diagonals in the sub-policies sequence illustrate the HC8 has learnt to chain its restricted sub-policies to learn a well-performing policy. We then combine this visualization with an exact description of which feature the chosen sub-policy used to interact with the environment, and with which magnitude (right part of Fig. 3, corresponding to the low-level interpretability of HC). In contrast, we emphasize that these visualizations do not bear any meaning for a regular actor. Indeed, SAC reuses the same linear coefficients in average 50 times over a 1000 timestep trajectory on Cheetah run, in spite of the cyclicity of the task. The information learnt by SAC is hidden by the interdependence of its sub-policies, preventing us from observing the emergent temporal abstraction phenomenon from Fig. 3 (see further details in Appendix C).

## 4.2 NAVIGATION

We now focus on maze navigation tasks (Nasiriany et al., 2019). We want to discover if (1) HC actors can solve hard exploration mazes by being combined with dedicated algorithms in spite of the restricted architecture and if (2) using HC actors in mazes can elicit temporal abstractions in the sub-policy sequence, similar to the control setting. The agent, a quadruped which state is described by proprioceptive features, is provided with a goal state to reach. The sparse reward signal makes the mazes hard to explore and require dedicated algorithms. In this context, we extend RIS (Chane-Sane et al., 2021), a goal-conditioned DRL algorithm. Given a state and goal, RIS encourages during training the policy to match the action predicted by a "prior" policy (an empirical moving average of the actor) to reach an intermediate goal. We replace the neural policy of RIS with HC actors with 8, 16 and 64 sub-policies, and otherwise follow the same training and evaluation protocol, which we detail in Appendix E. All HC actors use a $[64, 64, 64]$ architecture and use several regularization techniques. Compared to RIS, HC actors are trained with a reduced batch size (to accommodate computing resources) and use a stronger prior regularization hyperparameter (denoted $\alpha$).

**Performance** We observe in Fig. 4 that RIS as well as HC64 perform well in all the mazes: the constrained expressivity of HC actors does not prevent them from being trained on mazes using a dedicated navigation algorithm. While both RIS and HC64 get a stable performance for the U and S-shaped mazes, the wider confidence intervals for the other two mazes illustrate the difficulty of the tasks. Furthermore, HC8 and HC16 solve most of the time the U-shaped maze, and HC16 the S-shaped maze. All reach the goal in at least one of the seeds, showing that such architectures are not too constrained to express a goal-reaching policy. However, all HC actors require more interactions than RIS to solve the respective mazes, and the failure rate is overall higher. Compared to DM Control, the higher difficulty of the task (and of the assignation function needed to solve it) is one possible explanation to the reduced sample efficiency. Solving the maze requires high-level navigation in addition to low-level control, whereas only the latter was previously required.

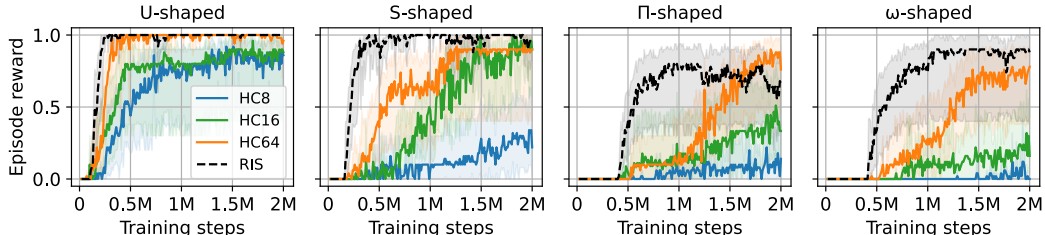

Figure 4: HC actors learn to solve the navigation tasks with a reduced sample efficiency.

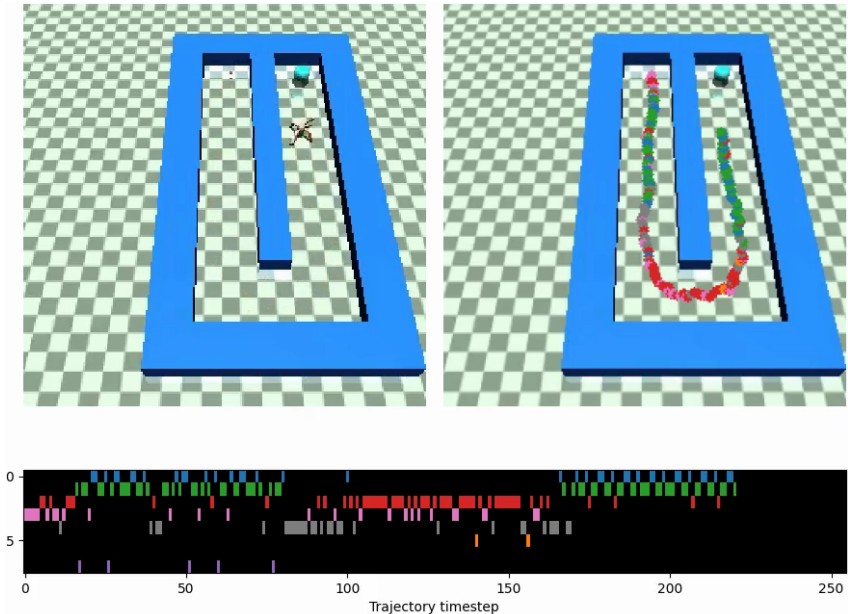

Figure 5: (Top left) The quadruped must go down the left corridor, and up the right corridor. (Bottom) HC8 sub-policy sequence after 2M steps. We note the appearance of motifs (repetitions of subsequences of sub-policies) that match the behavior observed on the agent: rotation, then straight movement, before a rotation to change corridor and a straight movement. (Top right) Trajectory of sub-policy choices superposed with the maze: each colored point refers to the sub-policy chosen at that location. The grouping of colors illustrate the empirical emergence of temporal abstractions.

**Interpretability** We finally provide an example of a sequence of sub-policies followed by HC8 during one successful trajectory, in Fig. 5. This visualization emphasizes how a temporal abstraction emerges from the hierarchical nature of HC. HC learns to alternate between sub-policies 0 and 1 to move straight, and between 2 and 3 to rotate, illustrating how the high-level interpretability of HC lets us identify behaviors followed by the agent to solve the maze. For instance, the usage of sub-policies 2 and 3 for the first steps indicates that the agent first needed to turn. A comparison with a video of the agent highlights that it was initialized facing the wrong direction, justifying this first rotation. We analyze a wider set of sub-policy sequences in other mazes in Appendix E.

## 5 RELATED WORKS

**Interpretability in (deep) reinforcement learning** Closest to our work, (Akrour et al., 2018) explore a very similar piecewise-linear policy architecture in the control setting, though not in the scope of interpretability. In particular, the choice of sub-policy is determined by a softmax function, such that the partition of the state space induced by the sub-policy assignation function is soft. On the contrary, the HyperCombinator induces a hard partition which guarantees that only a single sub-policy

is used at any interaction with the environment. Previous work also showed that linear (Rajeswaran et al., 2017) and discrete (Tang & Agrawal, 2020) policies performed surprisingly well in control environments, at the cost of a reduced flexibility and therefore a likely reduced ability to solve complex tasks. LMUT (Liu et al., 2019) generates functions that live in the same space as HC. However, LMUT is applied to the Q-function rather than the policy, is introduced in a distillation setting, and the partition it induces, while interpretable, is not learnt by gradient descent. Programmatically interpretable reinforcement learning algorithms (PiRL) train highly structured policies such as decision trees or finite state machines (Koul et al., 2018). Verma et al. (2018) and Bastani et al. (2018) distill an expert policy into an inherently interpretable one. Trivedi et al. (2021); Qiu & Zhu (2021); Liu et al. (2023) use policies defined over discrete actions (either from the environment or learnt) to solve high-level planning. On the contrary, HC aims to learn functionally interpretable policies that handle both to low-level continuous control and high-level planning. Contemporary to HC, Orfanos & Lelis (2023) learn an interpretable tree policy derived from a small actor. Other methods use non gradient-descent optimization methods to learn inherently interpretable policies (Hein et al., 2017; 2018). In high dimensional domains, the network can be intertwined with prototypes to justify inherently its decisions (Kenny et al., 2023). An attention module was added to the IMPALA algorithm, highlighting the parts of the image used for decision making (Mott et al., 2019). In our approach, we focus on proprioceptive states, and rather propose a policy with transparent interactions with the environment. Further from our work, there has been efforts to make the Q-function interpretable (Annasamy & Sycara, 2019; Juozapaitis et al., 2019). Tangentially, a recent work studied the evolution of the linear regions of a ReLU MLP in DRL (Cohan et al., 2022).

**Hierarchical methods**  Several navigation methods incorporate a hierarchical component in the policy training, which can inform the policy architecture (Nasiriany et al., 2019; Nachum et al., 2018; Faust et al., 2018) or guide it during training (Chane-Sane et al., 2021). We focus on making the policy more interpretable by building off the later method, restricting its expressivity. Options (Sutton et al., 1999; Bacon et al., 2017) are another type of temporal abstraction. In comparison, our method does not define an initiation or termination set, instead belonging to the usual RL formalism, with a simple policy architecture change. Close to our work, SENN (Alvarez Melis & Jaakkola, 2018) uses a similar architecture to HC, but focuses on supervised learning and adds a penalty to smooth the linear coefficients, while we structurally enforce the latter to be piecewise-constant and analyze the effects of the gained piecewise-linearity on the policy interpretability in RL. Finally, hierarchy can manifest directly in the network weights. Our architecture can be thought of as a hypernetwork (Ha et al., 2017) predicting the coefficients of a policy linear *w.r.t.* interpretable features.

## 6  DISCUSSION

In this article, we studied whether piecewise-linear policies can be a suitable compromise between performance and interpretability in DRL. To this end, we presented the HyperCombinator architecture, an implementation of a piecewise-linear actor that combines a controllably small set of inherently interpretable sub-policies to interact with the environment in a transparent manner. We showed that in several control and navigation environments, the performance loss due to the reduced expressivity of our policy class was limited. Moreover, we illustrated how new visualizations, possible thanks to the two levels of interpretability provided by HC policies, give us insights on the interactions of an agent with its environment. We also noticed the empirical emergence of temporal abstractions in the policy in both the control and select navigation settings.

This work could be improved in several ways. Currently, the assignation function learnt by the policy is not interpretable, which prevents explaining why a sub-policy was chosen. Moreover, the HyperCombinator interacts transparently with the environment, but is not a causal architecture. And so our policy ultimately does not fully answer "why" an action was taken, instead giving an interpretable decision conditioned on the sub-policy choice. Nonetheless, this is a significant step compared to current DRL algorithms, which are far from meeting the most basic standards of explainability.

The growing presence of AI models in our daily lives has raised the stakes about the abilities they should exhibit. In the face of an increasing need for interpretability, both by citizens (Vasconcelos et al., 2023) and by lawmakers, it is imperative to develop models that can accurately justify their decisions. Through the HC architecture, we showed that appropriately parametrized piecewise-linear policies can be an important first step towards meeting this expectation for DRL agents.

ACKNOWLEDGEMENTS

This work was supported by the Fonds de Recherche du Québec - Nature et Technologies (FRQNT), the Natural Sciences and Engineering Research Council of Canada (NSERC) and the Canada CIFAR AI chairs program. The authors thank Pascal Vincent, Alessandro Sordoni, Harsh Satija, Clara Lacroce, Jonathan Lebensold, Samuel Lavoie and Emmanuel Bengio for fruitful discussions, as well as colleagues from FAIR Montréal and Mila and anonymous reviewers for insightful comments and suggestions on earlier drafts of this document. Finally, we thank the Python (Van Rossum & Drake Jr, 1995), numpy (Van Der Walt et al., 2011), matplotlib (Hunter, 2007) and PyTorch (Paszke et al., 2017) communities that developed the tools this work is built on.

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

## A    Post-hoc, distillation and functional interpretability

We begin by setting the context of interpretability in machine learning. We wish to explain a *deployed model* and its predictions to users of varying expertise. There exists three main approaches to do so:

- A *post-hoc* method is an auxiliary model that extracts explanations from a complex (potentially a black box), well-performing deployed model (Ribeiro et al., 2016; Kim et al., 2016; Wachter et al., 2017; Sundararajan et al., 2017; Mothilal et al., 2020). We can expect good performance from the deployed model. However, the explanations are provided by the auxiliary model, which raises questions about their reliability, their accuracy or their robustness (Alvarez-Melis & Jaakkola, 2018; Rudin, 2019; Slack et al., 2020; 2021).

- *Distillation* methods first train a complex "teacher" model, and then fit and deploy a simpler "student" model on the predictions of the teacher (Hinton et al., 2015; Liu et al., 2019). The explanations come from the interpretable nature of the student (*e.g.* a shallow decision tree or a linear model). Yet, defining the data distribution to fit the student on can be challenging, especially in the DRL setting (Fujimoto et al., 2019; Ostrovski et al., 2021).

- *Functional* methods (Doshi-Velez & Kim, 2017) directly train an interpretable model, which is deployed and explain its decisions (Rudin, 2019; Kenny et al., 2023). The end-to-end approach to interpretability often imposes limits on expressivity that can imply a performance trade-off.

In this work, we focus on functional interpretability. To achieve it, one can reduce the size of the class that the model belongs to (Rudin, 2019). Classes of functions that can be expressed through a small number of parameters, such as linear functions or shallow decision trees, are often considered to be interpretable due to the reduced amount of information required to understand the function they realize (Molnar et al., 2021). Yet, designing such a model is challenging, since inherently interpretable models are at odds with the complexity and expressivity required to solve hard tasks. For instance, a linear policy might not be expressive enough to solve an intricate maze. Moreover, the inclusion of discrete components, while often involved in interpretable models (Doshi-Velez & Kim, 2017), can make training unstable.

## B    Reducing the size of the network to limit its complexity

### B.1    Theoretical analysis

Reducing the width or the height of an MLP is an alternative and more direct way than our proposed approach to reduce the number of unique coefficients. In fact, reducing the size of the MLP also reduces the size of the partition, *i.e.* the number of linear regions, on the contrary to our proposed solution. The main issue with this approach is that controlling the number of unique coefficients becomes impractical, since the maximum number of linear regions (which is the quantity we have control over) grows at least exponentially *w.r.t.* the depth and polynomially *w.r.t.* the width of the network (Montufar et al., 2014, Corollary 5). For instance, a relatively small MLP with a one-dimensional input and 2 layers of 64 neurons each can express a maximum of over 4000 linear regions (Montufar et al., 2014, Theorem 4). Finally, accessing the linear coefficient for a given feature $x_i$ requires a backward pass when using an MLP, while it can be directly outputted as part of our solution, since we compute this quantity explicitly in the forward pass.

### B.2    Empirical analysis

We complement the theoretical analysis of the upper bound by an empirical analysis of the actual number of unique linear regions used by small variants of the Soft Actor Critic algorithm (SAC). To do so, we train variants of SAC with an MLP actor architecture of depth 1, 2 and 3 and width of 2, 4, 8, 16, 32, 64, 128 and 256 units. We then roll out each variant in the Cheetah run environment (prone to sub-policy re-use due to the periodicity of the task) 10 times. We repeat the procedure for 10 seeds, which gives us, for each variant, 100 measures of the episode return (performance metric), and 100 number of unique sub-policies used (USPU) during the trajectory (complexity metric). We then produce a Pareto plot illustrating the trade-off between complexity and performance for small

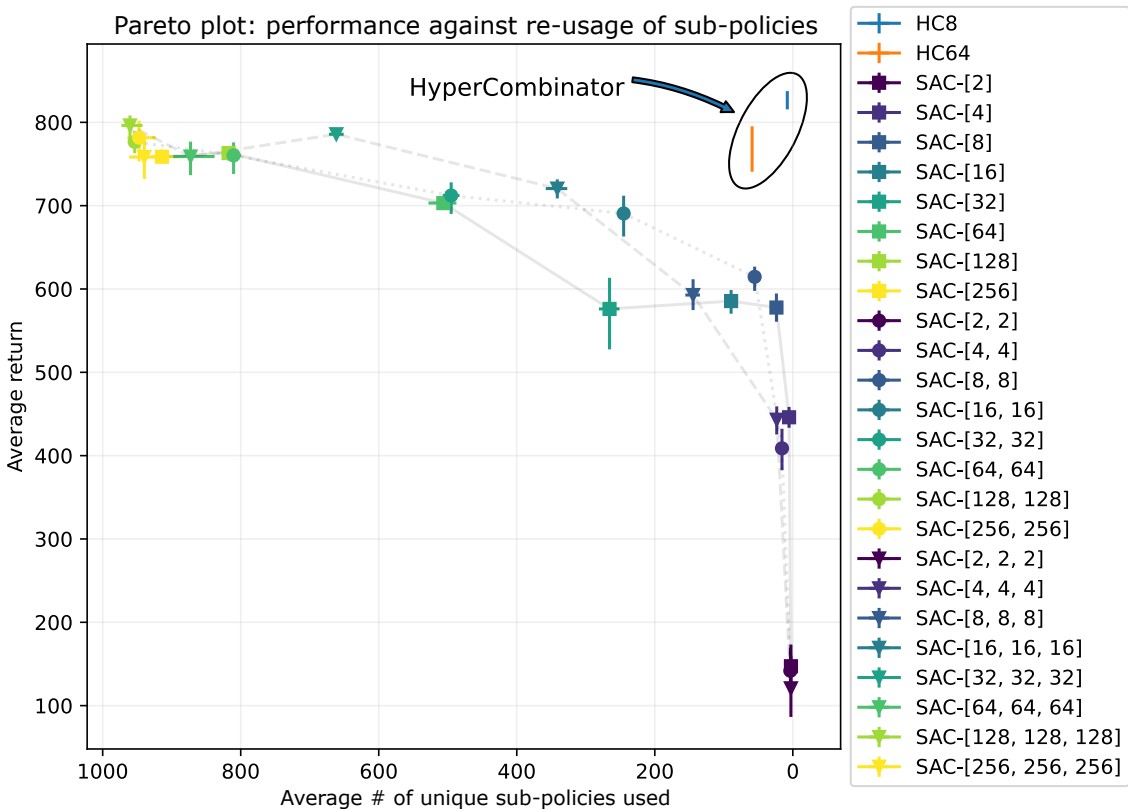

Figure 6: Pareto plot comparing the HyperCombinator actor and SAC with different small architectures on performance (y-axis, higher is better) and number of sub-policies used during a trajectory (x-axis, lower is better). Note the inversion of the x-axis, such that the best models are located in the top right part of the plot (with high return and low number of unique sub-policies used). We display error bars representing the 95% bias corrected and accelerated bootstrap confidence intervals (9999 resamples).

SAC architectures, in Fig. 6. We also compare the small variants of SAC with 2 variants of the HyperCombinator, respectively with 8 (HC8) and 64 (HC64) possible sub-policies.

We observe that overall, the performance of SAC is correlated with the number of unique sub-policies used during a trajectory. As a consequence, models with a small USPU metric tend to perform significantly worse than the more complex variants. For instance, to reach an average return of 700, variants of SAC need at least 200 unique sub-policies. This means that in average, each sub-policy is used only 5 times during a trajectory. In addition, if we limit the complexity to 100 USPU, the average return of SAC variants barely surpasses 600, well below the average return of 800 reached by the more complex SAC variants. Conversely, our proposed approach offers a better trade-off between complexity and performance SAC. Indeed, HC8 and HC64 are located further than the SAC Pareto front, as they manage to obtain significantly higher return for comparable complexity, or equivalently a comparable return with a much smaller number of unique sub-policies used. While in this particular example, HC8 outperforms even the SAC variants with the biggest architectures (see also Fig. 12), we insist specifically on the excellent performance of HC8 and HC64 *w.r.t.* SAC variants that use a comparable number of USPU.

In conclusion, simply limiting the architecture of SAC is not an adapted solution to our problem, since one cannot exactly control the number of unique sub-policies that will be used, and since the performance decreases severely for low number of unique sub-policies used ($<100$), even more so with very low number of unique sub-policies used ($<10$).

## B.3 L1 PENALTY

We consider adding an L1 penalty on the parameters of the actor. The goal is to encourage sparse policies, that would hopefully use less linear sub-policies within a trajectory. We consider various L1 penalties, as well as a small (1 layer of 4 units), medium (one layer of 32 units) and big (two layers of 1024 units) architectures. We illustrate the results in Table B.3. We observe that for the combinations that were tried, the L1 penalty was not sufficient to reach a satisfying performance-interpretability trade-off.

| architecture & penalty | episode reward (95% CI) | unique count (95% CI) |
|---|---|---|
| big, $\lambda$ = 1e-4 | 0.8 (0.7, 0.9) | 1.0 (1, 1) |
| big, $\lambda$ = 1e-5 | 814.8 (804.0, 824.7) | 917.4 (902.6, 928.9) |
| big, $\lambda$ = 1e-6 | 760.7 (724.5, 789.1) | 965.9 (948.5, 978.5) |
| big, $\lambda$ = 1e-7 | 822.6 (782.2, 838.8) | 962.1 (940.4, 978.8) |
| med, $\lambda$ = 1e-1 | 0.1 (0.1, 0.1) | 1.0 (1, 1) |
| med, $\lambda$ = 1 | 0.1 (0.1, 0.1) | 1.0 (1, 1) |
| med, $\lambda$ = 2.5 | 0.1 (0.1, 0.1) | 1.0 (1, 1) |
| med, $\lambda$ = 10 | 0.1 (0.1, 0.1) | 1.0 (1, 1) |
| small, $\lambda$ = 1e-4 | 390.8 (368.1, 400.0) | 4.4 (3.9, 4.9) |
| small, $\lambda$ = 1e-5 | 428.7 (413.2, 445.4) | 9.7 (8.8, 10.5) |
| small, $\lambda$ = 1e-6 | 417.4 (393.7, 436.8) | 5.6 (4.9, 6.3) |
| small, $\lambda$ = 1e-7 | 384.7 (338.3, 425.5) | 4.5 (4.2, 4.9) |

## B.4 DISTILLATION

An alternative to learning a small policy with reinforcement learning is to first train a complex policy with reinforcement learning (a teacher) and then fit in a supervised learning fashiong a small policy (a student) to imitate the actions predicted by the teacher. We now compare the HyperCombinator to such a strategy in the Cheetah run task.

We first train to convergence a SAC agent with two hidden layers of 1024 units, following the baseline hyperparameters detailed in D.1. We repeat the experiment over 3 seeds, and select the seed with the highest episode return averaged over 10 rollouts. The corresponding agent is referred to as the (fully trained) teacher. We save the replay buffer, which contains the sequence of observations that the teacher encountered over training. We then label each observation with the action predicted by the fully trained teacher. This methodology yields a dataset of 1M observations and the corresponding ideal action.

We now focus on training the student on this dataset. We minimize the mean squared error between the student predicted action and the teacher predicted action for each observation in the dataset. We detail the hyperparameters in Table 1. In addition, we compare results for different batch sizes (256 and 1024) and l1 penalty coefficients ($\lambda_{L1} \in \{1e-6, 1e-5, 1e-4, 1e-3\}$).

| Hyperparameter name | Value |
|---|---|
| Student actor architecture | [32] |
| Optimizer | Adam |
| Learning rate | 1e-3 |

Table 1: List of hyperparameters in the distillation experiments.

Performance is evaluated several times during training by rolling out the student in the Cheetah run task and recording the rollout return. We repeat this operation 10 times and compute the average rollout return. We repeat the experiment over 5 different seed and average the results, and compute 95% bootstrap confidence intervals. We illustrate this curve in Fig. 7, where we also plot the return curve for HC8. We observe that all the different students failed at getting an acceptable performance in the task. We observe that no combination of hyperparameter led to a well-performing student in

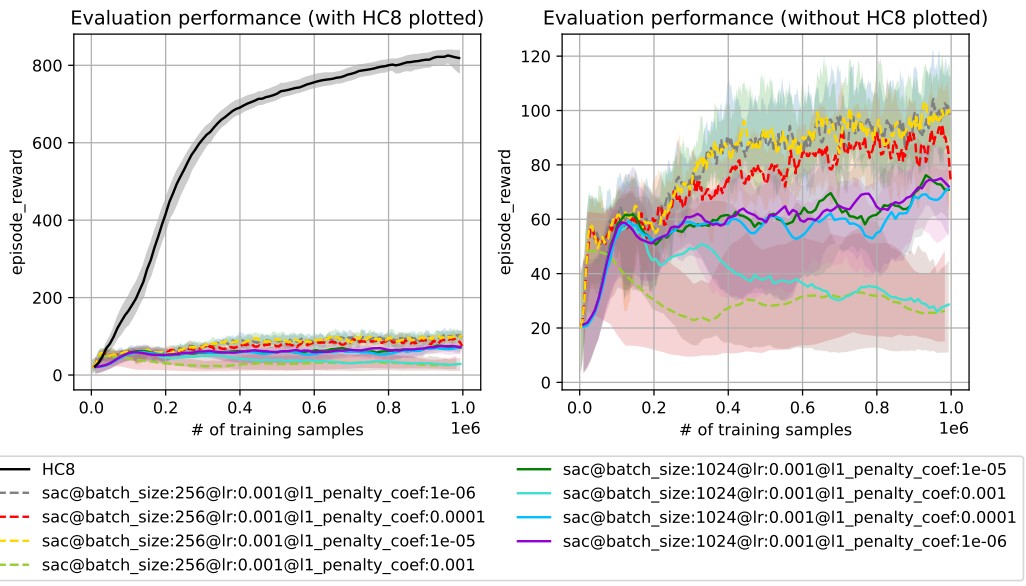

Figure 7: Students trained to imitate the teacher fail catastrophically.

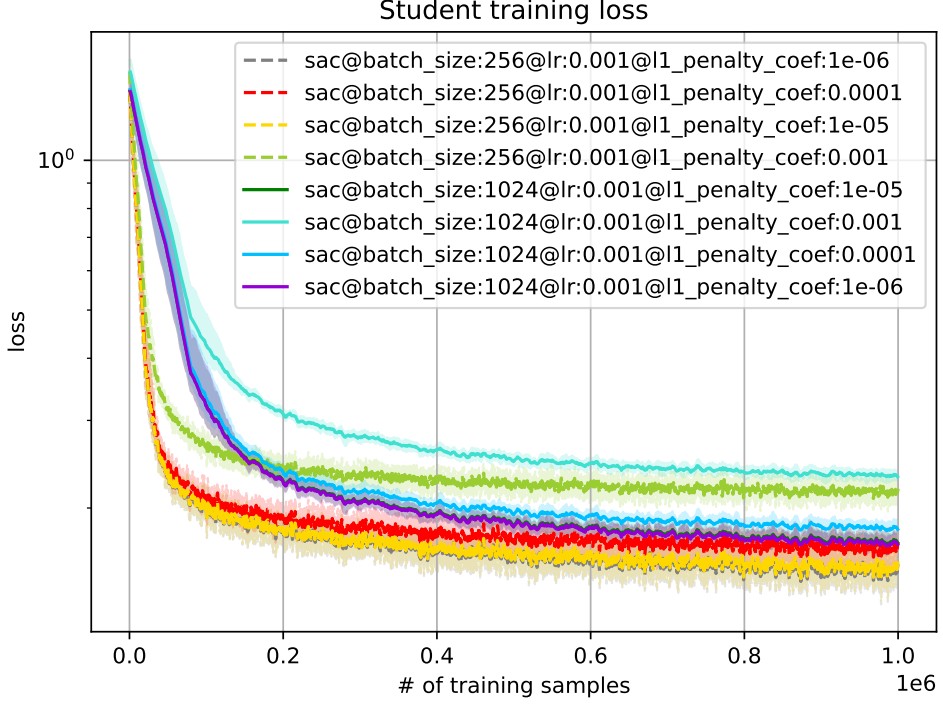

Figure 8: Training loss curves for each student combination.

this task. Students that were distilled from the teacher did not manage to learn to solve the task. In addition, high L1 penalty coefficient degraded performance.

We also plot in log scale the training curves in Fig. 8, to illustrate that the students do learn to minimize the error with the teacher predictions. The lower capacity of the students, which forces them to make prediction errors at potential key states in the trajectory, is a possible reason for their catastrophic performance.

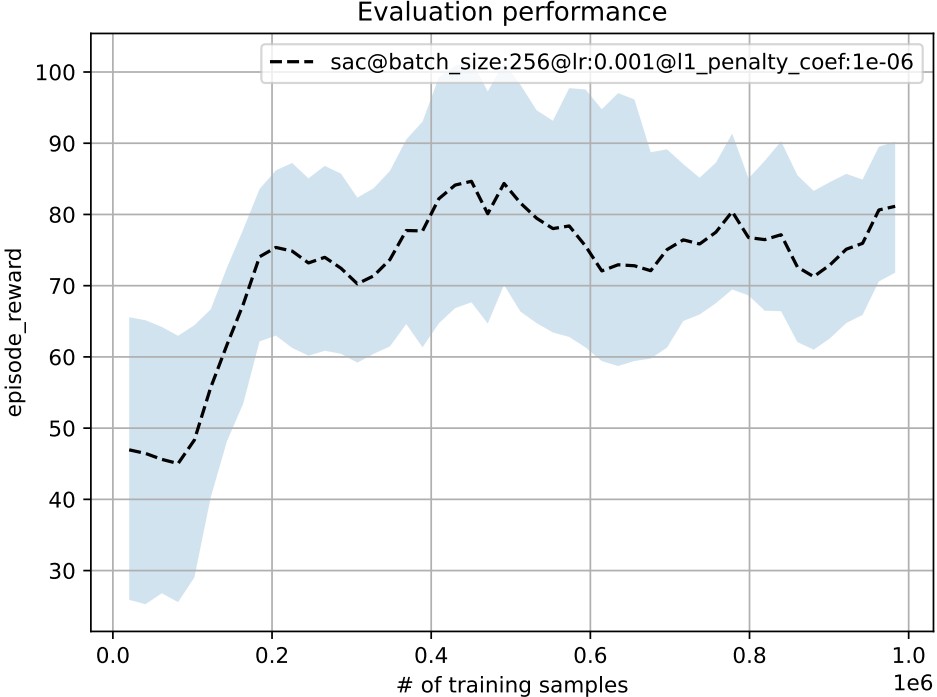

Figure 9: Input augmentation does not improve the performance of the agent.

We also consider training periodically the student on inputs that it generates itself, which is referred to "input augmentation" in Verma et al. (2018). We evaluate the student every 80 student updates (corresponding to 20480 data points with a batch size of 256). The 10 evaluation rollouts generate each 1000 observations, leading to 10000 inputs that were generated by the student interacting with the environment. We train the student on these inputs at the end of the 10 evaluation rollouts, and then resume training on the teacher data. We show in Fig. 9 that input augmentation does not improve the performance of the student in our case.

Finally, we presented one way to distill a teacher into a student network. We note that there are other ways we could implement distillation algorithms, for instance by mixing on-policy distillation and a reinforcement learning loss (Schmitt et al., 2018), or adding a feature regression objective to the reinforcement learning loss (Parisotto et al., 2015). All of these methods add an additional inductive bias compared to our algorithm. One could also modify the training data, for instance by using a more exploratory expert policy to gather a wider range of data points.

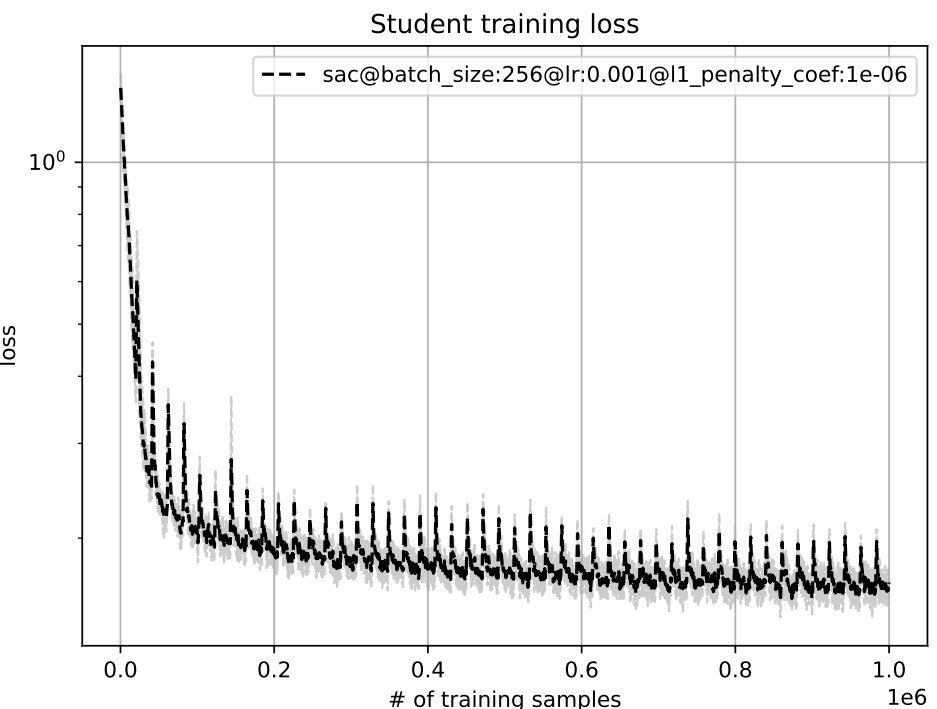

Figure 10: Training loss curve with input augmentation. We hypothesize the spikes are due to training the low-capacity student on the data it generated, which differs significantly from the data generated by the teacher.

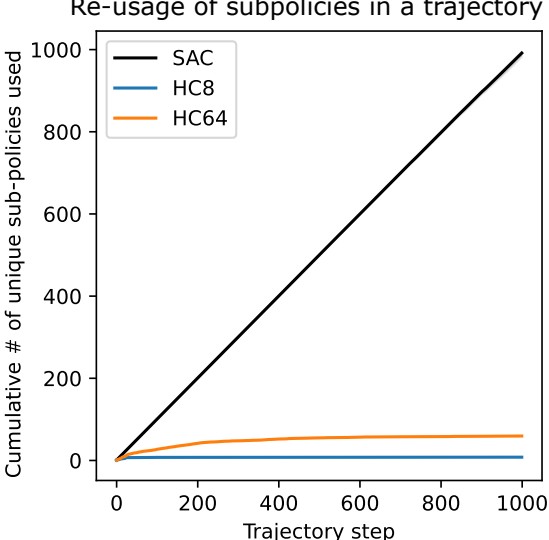

Figure 11: During a trajectory, a SAC actor visits a new linear region (and uses a new locally linear sub-policy) at almost every timestep. On the contrary, our HC agent can only express one of the 8 or 64 sub-policies at his disposal, which forces it to learn sub-policies that are valid for a wide range of inputs.

## C   SUB-POLICIES REUSE

Since SAC uses an MLP as the policy, the policy is also a piecewise-linear function. We study in this section how often does the policy re-use the same linear region in a trajectory.

We perform the experiment using the trained models of Sec. 4.1, on the Cheetah run environment, which is particularly favorable to the reuse of sub-policies. We rollout the trained policies of SAC, HC8 and HC64 for an entire trajectory and record, for each timestep, the number of unique sub-policies used to interact with the environment until the current timestep. This yields a non-decreasing sequence of integers per algorithm. The lower the curve, and the less sub-policies are being used by the policy. We repeat the same procedure 10 times, and compute the mean curve as well as a 95% confidence interval using the bias corrected and accelerated bootstrap (9999 resamples).

We display the results in Fig. 11. The HyperCombinator cannot express more than $d_G$ sub-policies, which caps the corresponding plots for both HC8 and HC64. We observe that the curve corresponding to SAC linearly increases almost without failure. This indicates that SAC barely reuses past sub-policies, *i.e.* that a new locally linear function is applied at practically each timestep. Moreover, this sub-policy is shared by all seeds, given the thinness of the confidence interval.

This does not mean that SAC strictly overfitted the trajectory or completely memorized the sequence of policies to apply during the trajectory: indeed, there is a great amount of parameter sharing between the linear coefficients in $\theta$. Thus, there is generalization happening between sub-policies, even if a different linear region is used at each timestep. Yet, the precise generalization mechanism is obscure, and it is not clear to what extent updating a sub-policy will affect the other sub-policies. Perhaps more importantly to our use case, one still needs to enumerate all of the sub-policies when describing the possible interactions of the policy with the environment, given that even though they might share parameters, each sub-policy uses a different set of linear coefficients. As a consequence, this makes the SAC policy highly uninterpretable, on the contrary to the HyperCombinator where the size of the exhaustive list of sub-policies is capped by $d_G$.

# D  CONTROL EXPERIMENTS

## D.1  IMPLEMENTATION DETAILS

We base ourselves on an open-source PyTorch implementation of SAC (Yarats & Kostrikov, 2020) with its default hyperparameters, listed in Table 3. We detail below the modifications needed to implement the HyperCombinator and replicate the experiments, and keep the rest of the existing code for training and evaluating SAC and HC.

**SAC algorithm**  We modify the actor of the Soft Actor Critic algorithm (SAC) (Haarnoja et al., 2018), a strong continuous control agent learning from proprioceptive observations. The original actor is modeled as an MLP that predicts a vector of size $2|\mathcal{A}|$. The output vector is then split into two parts, the mean of the action predictive distribution, $\tilde{\pi}$ and its log standard deviation, $\log \sigma$. We note that this implies that the mean $\tilde{\pi}$ and the log std $\log \sigma$ share some of their parameters through the shared structure of the actor. The log standard deviation is transformed to belong to the interval $[\log \sigma_{\min}, \log \sigma_{\max}]$ using the following formula:

$$t(u) = \log \sigma_{\min} + (\log \sigma_{\max} - \log \sigma_{\min}) \cdot \frac{u+1}{2} \tag{4}$$

Finally, the action predictive distribution for state $x$ is defined as a *squashed normal*:

$$\mu \left( \mathcal{N}(\tilde{\pi}(x), \exp t(\log \sigma(x))) \right) \tag{5}$$

where $\mu$ is the non-linear transformation ensuring that the action belongs to the environment action space. In practice for the DeepMind Control Suite, we use $\tanh$ as $\mu$ to force the predicted actions to belong to $[-1, 1]$.

The critic is learnt via double Q learning (Van Hasselt et al., 2016). Each critic network is a 2-layer MLP. The strength of SAC entropy regularization (represented by the hyperparameter $\alpha$) is automatically learnt during training. We use Adam to optimize all parameters. All weight matrices are initialized using orthogonal initialization. All bias vectors are initialized to 0.

**HyperCombinator modifications**  With the HyperCombinator, we model the mean $\tilde{\pi}$ and the log std $\log \sigma$ with independent networks. HC first models $\log \sigma$ as an MLP with the same architecture as in SAC, but that only outputs a vector of size $|\mathcal{A}|$. Then, HC models $\tilde{\pi}$ as a Gumbel network, that is, the composition of an MLP, a Gumbel-Softmax layer (with the straight-through estimator) and a linear layer. The rest of the computation of the action predictive distribution is left unchanged. In particular, for $x$, the mean of the squashed normal is in both cases $\mu(\tilde{\pi}(x))$ and does not depend on $\log \sigma(x)$.

We detail the modified SAC algorithm using Alg. 1. We indicate in **blue** the modifications due to using a HyperCombinator actor. We remark that the only major modification is in the architecture of the actor. Therefore, the majority of the structure of the SAC algorithm is left unchanged. We notice the use of stop_grad to detach a tensor from the computation graph.

**Training details**  Classically, Gumbel noise is used in the Gumbel-Softmax layer to stochastically select the sub-policy. After sufficient training, this (Gumbel) trick ensures that we ultimately sample from the likeliest sub-policy. We only activate Gumbel noise during the actor update, when computing the action predicted by the actor. When the agent interacts with the environment, we do not use Gumbel noise, both for training and for evaluation. Therefore, the Gumbel network selects the most likely *sub-policy* for all interactions with the environment. This ensures that the *mean* of the action predictive distribution is deterministic, like SAC. It also improves the consistency of the agent over an episode, since for state $x$, a fixed policy leads to the same sub-policy. This is not to be confused with the action predicted by the HyperCombinator, which is sampled from the squashed normal defined in Eq. 5 during training.

**Model choice**  We experiment with different combination of hyperparameters values (Gumbel network architecture from [64,64] to [1024,1024,1024], and strength of regularization of the average assignment entropy $\lambda_{\text{assig}}$ from 0 to 1) on Cheetah run, from where we selected the best set of hyperparameters according to the return curve. We then evaluate the HyperCombinator using the same fixed set of hyperparameters for all the environments.

**Evaluation details**    We evaluate the agent every 10000 timesteps by rolling it out for 10 episodes and taking the average return. During evaluation, all actors act deterministically using only the mean of the predicted action distribution $\mu(\tilde{\pi}(x))$, without exploration noise (as opposed to sampling from the predictive action distribution during training). Hence, it is sufficient for $\mu(\tilde{\pi})$ to be piecewise-linear to get the desired form for the actor at evaluation time. We report all results and curves in Sec. 4.1 using 10 seeds for each agent. We draw the mean performance as a colored line, as well as a 95% bias-corrected and accelerated bootstrap confidence interval in a lighter shade (9999 resamples).

**Compute**    We ran all the experiments on an internal cluster. All the GPUs were NVIDIA Tesla V100, with 16GB memory available. The CPUs were Intel(R) Xeon(R) CPU E5-2698 v4 @ 2.20GHz Each seed was allocated 1 GPU, 10 CPUs, and 64GB of RAM. We detail the compute budget to reproduce the experiments in Table. 2.

| Experiment | # models | # envs | # seeds | Avg. duration | Compute |
|---|---|---|---|---|---|
| Full results (Fig. 12) | 3 | 23 | 10 | 6 hours | 173 GPU days |
| Longer horizon (Fig. 16) | 3 | 4 | 10 | 34 hours | 170 GPU days |
| Perf.-interpret. gap (Fig. 17) | 8 | 1 | 10 | 17 hours | 57 GPU days |
| Small SAC (Fig. 6) | 24 (*) | 1 | 10 | 6 hours | 60 GPU days |

Table 2: Compute budget for the control experiments. (*) the HyperCombinator plots were already computed in the full results.

| Hyperparameter name | Value |
|---|---|
| *Common (SAC defaults)* (Yarats & Kostrikov, 2020) | |
| Action repeat | 1 |
| Discount factor | 0.99 |
| Learnable $\alpha$ | True |
| Initial $\alpha$ | 0.1 |
| $\alpha$ learning rate $\lambda_\alpha$ | 1e-4 |
| $\alpha$ Adam momentums | [0.9, 0.999] |
| Actor learning rate $\lambda_\pi$ | 1e-4 |
| Actor Adam momentums | [0.9, 0.999] |
| Actor update frequency | 1 |
| Critic architecture | [1024, 1024] |
| Critic learning rate $\lambda_Q$ | 1e-4 |
| Critic Adam momentums | [0.9, 0.999] |
| Critic exponential moving average ratio | 0.005 |
| Critic target update frequency | 2 |
| Batch size | 1024 |
| $\log \sigma_{\min}$ | -5 |
| $\log \sigma_{\max}$ | 2 |
| *SAC actor-specific* | |
| Actor architecture[1] | [1024,1024] |
| *HyperCombinator-specific* | |
| Gumbel net architecture | [1024, 1024, 1024] |
| Sub-policy assignation entropy coefficient $\lambda_{\text{assig}}$ | 0.001 |
| Gumbel temperature | 1 |

Table 3: Full list of hyperparameters in the control experiments.

---

[1]Shared between the mean net and the log std net

---

**Algorithm 1** `SAC (with HyperCombinator actor)`

---

**Require:** Replay Buffer $\mathcal{D}$
**Require:** Actor parameters $\varphi$
**Require:** Double critic parameters $\eta_1, \eta_2$
**Require:** Double critic target parameters $\overline{\eta}_1, \overline{\eta}_2$
**Require:** Hyperparameters from Table 3
**Require:** $N$               $\triangleright$ Maximum number of timesteps
**Require:** $s_0$                 $\triangleright$ Initial state
 **while** $t < N$ **do**
  $a_t \sim \mu(\tilde{\pi}(s_t))$
  $s_{t+1} \sim P(\cdot|s_t, a_t)$       $\triangleright$ Sample the next state from the environment
  $r_{t+1} = R(s_t, a_t, s_{t+1})$
  $\mathcal{D} = \mathcal{D} \cup (s_t, a_t, r_{t+1}, s_{t+1}, d_{t+1})$     $\triangleright$ Update replay buffer; $d_{t+1}$ indicates a terminal
 transition

  $(s, a, r, s', d) \sim \mathcal{D}$          $\triangleright$ Sample from replay buffer
  Launch routine `UpdateCritic`     $\triangleright$ See paper (Haarnoja et al., 2018) and code[2]
  **if** $t$ % actor update frequency $== 0$ **then**
   Launch routine `UpdateActorAndAlpha` (see Alg. 2)
  **end if**
  **if** $t$ % critic target update frequency $== 0$ **then**
   $\overline{\eta}_1 = (1 - \text{critic ema ratio}) * \overline{\eta}_1 + \text{critic ema ratio} * \eta_1$
   $\overline{\eta}_2 = (1 - \text{critic ema ratio}) * \overline{\eta}_2 + \text{critic ema ratio} * \eta_2$
  **end if**
 **end while**

---

**Algorithm 2** `UpdateActorAndAlpha`

---

**Require:** $s$           $\triangleright$ Batch of states sampled from the replay buffer
 $a \sim \mu(\mathcal{N}(\tilde{\pi}(s), \exp(t(\log \sigma(s)))))$   $\triangleright$ $\tilde{\pi}$ is a Gumbel network and $\log \sigma$ an MLP instead of being jointly parametrized as an MLP in the base case.
 $\mathcal{L}_{\text{assig}} = H(\text{mean}(\hat{a}(s)))$      $\triangleright$ Compute entropy of average sub-policy assignation
 $Q = \min(Q_{\eta_1}(s, a), Q_{\eta_2}(s, a))$
 $\mathcal{L}_\pi = \text{mean}(\text{stop\_grad}(\alpha) \log \pi(a|s) - Q) - \lambda_{\text{assig}} \mathcal{L}_{\text{assig}}$ $\triangleright$ Do not backprop. gradients through $\alpha$
 $\varphi = \varphi - \lambda_\pi \nabla_\varphi \mathcal{L}_\pi$
 **if** learn $\alpha$ **then**
  $\mathcal{L}_\alpha = \text{mean}(\alpha \text{ stop\_grad}(\log \pi(a|s) + |\mathcal{A}|))$
  $\alpha = \alpha - \lambda_\alpha \nabla_\alpha \mathcal{L}_\alpha$
 **end if**

---

## D.2 RESULTS FOR ALL DEEPMIND CONTROL ENVIRONMENTS

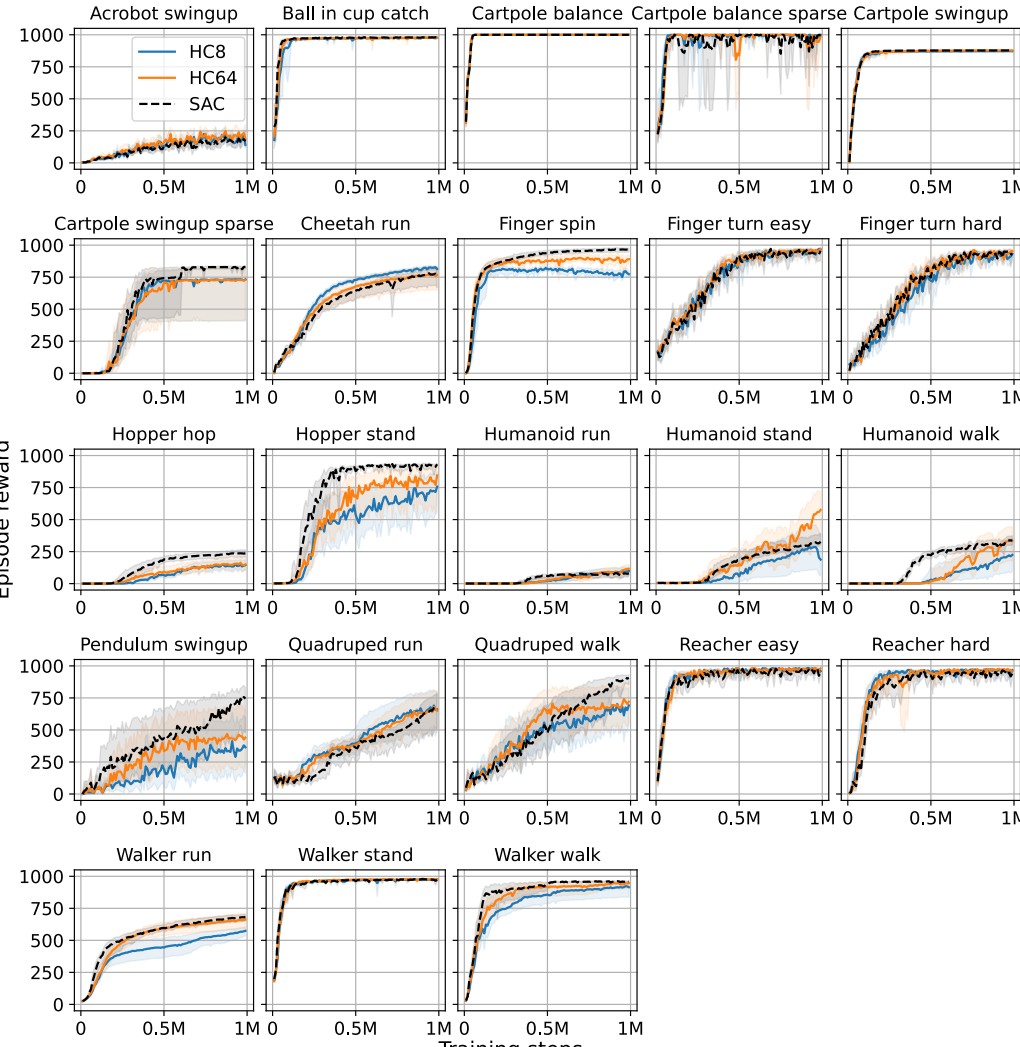

Figure 12: Complete results for the DM control environments.

## D.3 DETAILED QUANTITATIVE RESULTS

To compute the results, we followed the methodology proposed in the `rliable` library (Agarwal et al., 2021). We used the final scores after 990000 frames for each run.

We first investigate the scores produced by `rliable` in Fig. 13. This lets us quantify the trade-off in performance that follows the reduction in expressivity of HC8 and HC64 according to several metrics. In particular, the overlapping confidence intervals between SAC and HC64 indicate that overall, a very small amount of performance is foregone when one is willing to guarantee that the agent will interact with the environment in at most 64 different ways.

Fig. 14 illustrates the close performance between SAC and HC64, especially in the environments where the normalized score is at least 80% of the maximum (*i.e.*, in the games where SAC excels to begin with). HC8 follows a similar curve, albeit sensibly lower.

Finally, Fig. 15 shows the fraction of environments where the normalized score IQM of HC8 (respectively HC64) is at least a percentage of the baseline SAC score. This visualization is helpful

to distinguish the relative performance of the HC actors, compared to SAC, on the different environments. We remark that both for HC8 and HC64, there is a sudden drop around the 0% relative performance mark, which indicates that HC8 and HC64 have close IQM scores to SAC in several of the environments. Moreover, HC8 and HC64 perform worse than SAC on most environments by the end of training, as expected.

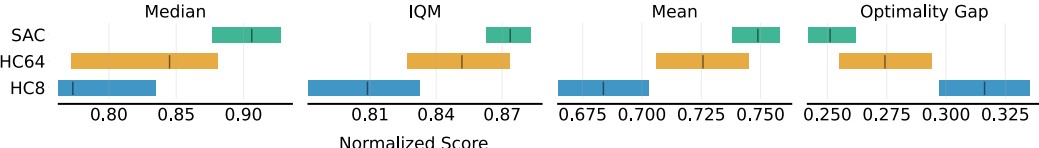

Figure 13: Performance metrics computed using the `rliable` library.

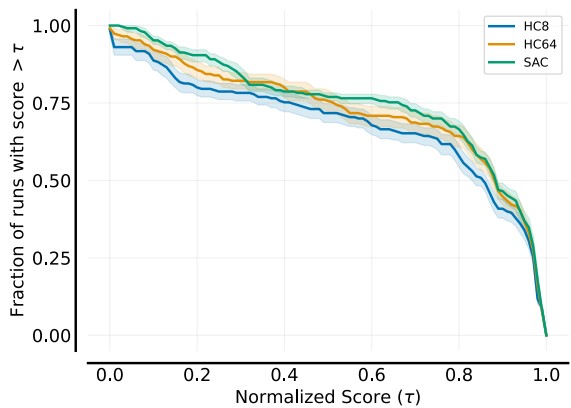

Figure 14: Performance profiles.

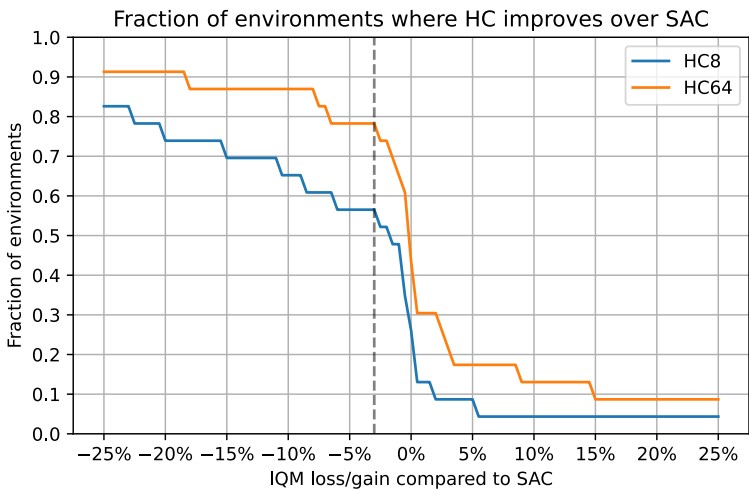

Figure 15: Relative performance loss/gain of the HC actors compared to SAC.

### D.4 LONGER HORIZON

In this section, we train the different actors for 5M steps instead of 1M as in the rest of Sec. 4.1. We focus our study on 4 environments: the first is Cheetah run, to observe if the benefits brought to early training by the usage of the HyperCombinator are also transferred to later during the training. The other 3 other environments, Humanoid stand, walk and run, are chosen for their increased difficulty. We illustrate the results in Fig. 16. We remark that HC actors improve over SAC for the first 1M timesteps in the Cheetah run environment. After approximately 1M timesteps, the performance of the SAC actor exceeds the HyperCombinator's. For Humanoid run and stand, both Hypercombinator improve over SAC approximately between 1M and 3M timesteps, with a significant improvement noted for Humanoid stand. For Humanoid walk, only HC64 improves over SAC, here again approximately between 1M and 3M timesteps. The competitive performance of the HyperCombinator actors, added to their improved performance in these hard environments early during training, hints at the benefits of sub-policies re-use. Still, we note that given sufficiently many samples, the more complex neural policies of SAC outperform HC. This recalls the trade-off between interpretability and performance observed in the long run, induced by choosing a lower complexity policy when using HC.

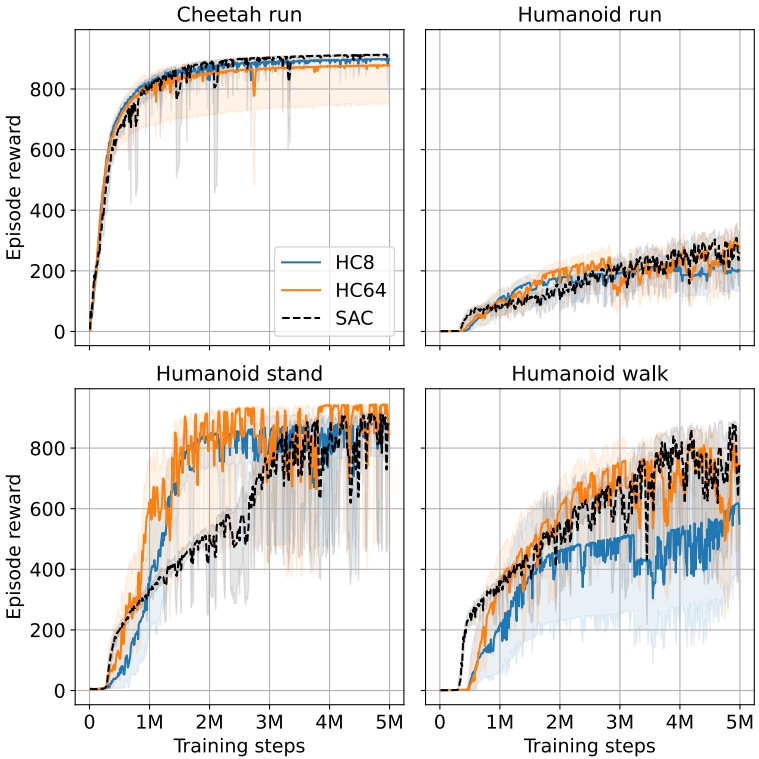

Figure 16: During early training, HC actors can demonstrate a better sample efficiency thanks to the re-use of the same sub-policy for several inputs. Over a longer timeframe, the higher expressivity of SAC leads to matching or better performance.

### D.5 PERFORMANCE-INTERPRETABILITY GAP

In this section, we evaluate how the HyperCombinator agent's performance evolves when increasing the number of sub-policies. We run a HyperCombinator actor with 4, 8, 16, 32, 64, 128 and 256 sub-policies on Walker walk for 2.5M timesteps, and observe the results in Fig. 17.

We remark that HC4 reaches a relatively high return, in average higher than 900. This shows that even with a small number of sub-policies, the HyperCombinator can learn a well performing policy in certain environments. As we trade-off interpretability for performance by increasing the number of sub-policies, we see the performance curve increase as well and approach the performance curve of SAC. This trend culminates with 128 sub-policies. We notice that the return curve of HC256 decreases compared to HC128.

A HC agent with a higher number of available sub-policies can produce a more finely grained policy, since each sub-policy can specialize to a smaller part of the state space. Conversely, learning how to chain a high number of sub-policies might not be straightforward. This last point is one possible explanation to the lower performance of HC256.

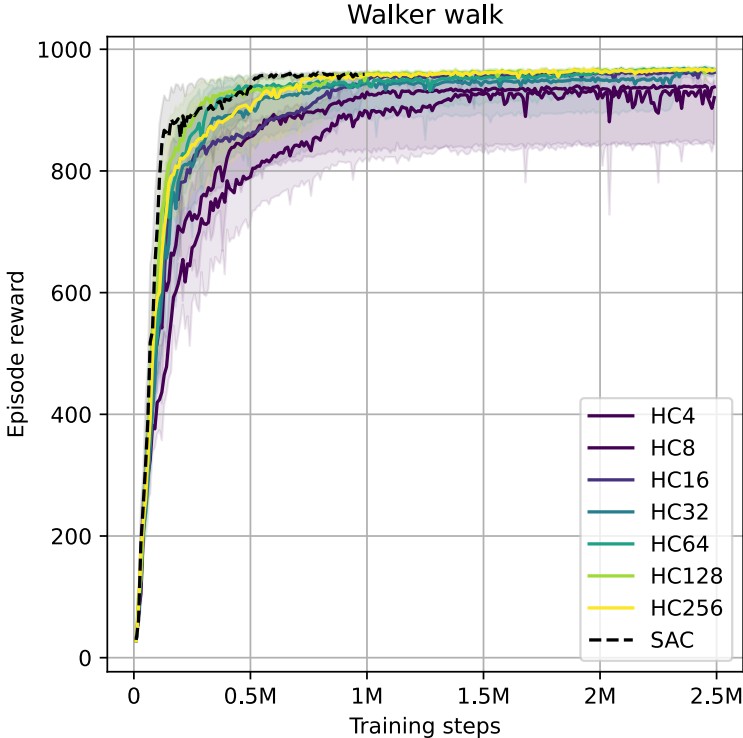

Figure 17: Increasing the capacity tends to increase the performance, as the policy is now able to exhibit a wider variety of sub-policies. A sufficiently high capacity can help reach a comparable performance to the base algorithm, here SAC. We notice that too many sub-policies can however prove harmful: the performance of HC256 sub-policies decreases compared to HC128.

### D.6 ROBUSTNESS TO PERTURBATIONS

In this section, we evaluate the robustness of the HyperCombinator to perturbations. We start with a regularly trained agent on the Cheetah run task. At evaluation time, after 100 timesteps (out of a 1000 timesteps trajectory), we ignore the action predicted by the agent and instead interact with the environment through an action sampled uniformly at random from the action space. We do so for 100 consecutive timesteps. At timestep 200, we resume normal evaluation and follow the action predicted by the agent. This perturbation throws the agent off its trajectory.

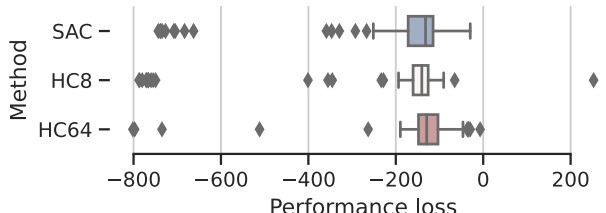

Figure 18: When perturbed with 100 consecutive random actions, HC actors perform overall no worse than SAC. The box plot summarizes the results of the experiment repeated 10 times for each seed.

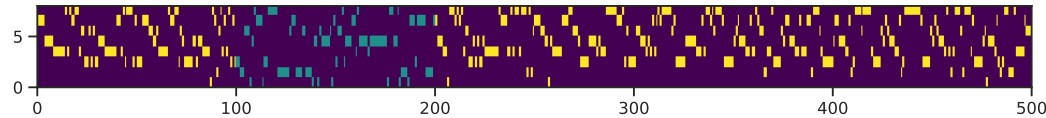

Figure 19: After being perturbed for 100 timesteps, the HC actor manages to resume the task. sub-policy choices between t=100 and 200 (in turquoise) are ignored since a random action is used instead.

We repeat this evaluation 10 times per seed, for each of the 10 seeds of an algorithm, for a total of 100 data points.

We now compare how this perturbation affects HC8 and HC64, compared to the baseline, SAC. In Fig. 18, we compute for each model the distribution of performance losses, defined as the performance of the perturbed evaluation subtracted to the performance of the unperturbed evaluation. We then illustrate the distribution of these performance losses. We see that all models react similarly to the perturbations. This illustrates the fact that the HyperCombinator is no less robust to perturbations than the base neural policy, despite its non-continuity. A similar conclusion was already found when comparing linear models to neural baselines (Rajeswaran et al., 2017). Accordingly, diversifying the initialization distribution could improve the robustness of the model.

We illustrate in Fig. 19 how the agent, after being perturbed for 100 timesteps, manages to resume the cyclical choice of sub-policies that lets it solve the task.

## D.7 ANALYSIS OF THE LINEAR COEFFICIENTS INTERPRETABILITY IN CARTPOLE

We now analyze the low-level interpretability benefits of our method when applied to the Cartpole environment.

In this environment, the agent controls the translation of a cart on an axis in order to swing up and then stabilize a pole. We write the sub-policy $i$ as: $b + w_x^i x + w_{\sin \theta}^i \sin \theta + w_{\cos \theta}^i \cos \theta + w_{\dot{x}}^i \dot{x} + w_{\dot{\theta}}^i \dot{\theta}$, for an input $(x, \sin(\theta), \cos(\theta), \dot{x}, \dot{\theta})$ and sub-policy coefficients $w^i$. Note that there is a single action: for a given positive feature, the contribution tends to move the cart towards the right if the corresponding coefficient is positive and left if negative.

We visualize the results in Fig. 20. The left plot is an alternative visualization of the sub-policy coefficients $w_i$ (to be compared with the heatmap in Fig. 3, right). This is an easy way to note similarities between sub-policies and to understand how they are combined. For instance, HC mainly uses sub-policy B0 (in orange) to stabilize the pole that has been swung up. This is translated by a negative coefficient in front of the cart velocity, such that if the agent moves towards the right (causing the pole to tilting to the left), the sub-policy will push the agent towards the left to balance the pole.

We also see in Fig. 20 (right) that the agent sometimes oscillates between using B0 (orange) and sub-policy 1 (grey). We can observe in the left plot that they both give a large coefficient to $\cos \theta$, which is around 0 when the pole is stabilized. The positive sign of the coefficient indicates that if the pole

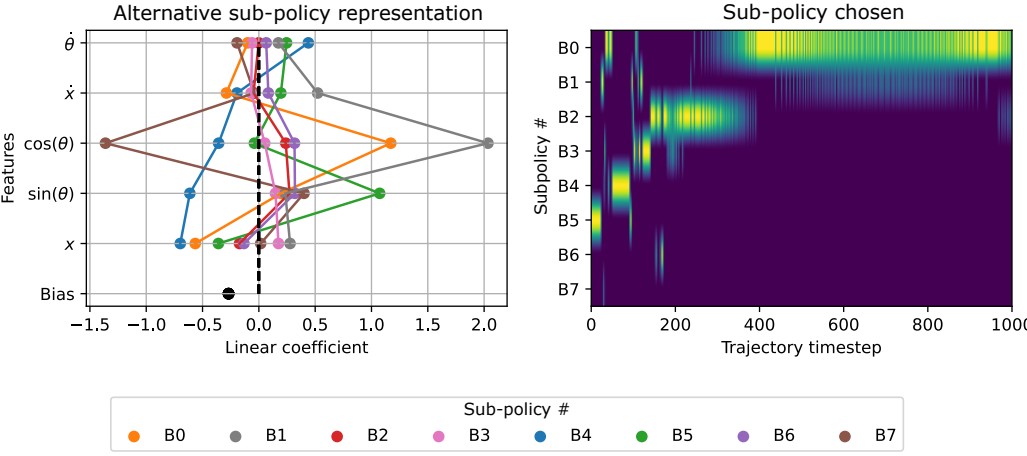

Figure 20: Evaluation of HC8 in cartpole swingup. (Left) alternative visualization of the sub-policies. (Right) sub-policy choice sequence.

is tilting towards the right, the agent will tend to move the cart towards the right to balance the pole. Finally, Fig. 20 (left) shows that B2 (red) and B6 (purple) are extremely similar. Simultaneously, we see in Fig. 20 (right) that the only setting in which B6 is used is during an extended period of usage of B2.

We think that this figure is an example of how using a HC actor (rather than a neural one) can improve the interpretability and our understanding of the task.

### D.8 COMPARISON WITH PROGRAMMATIC POLICIES

We compare in this section the HC architecture with the programmatic policy presented in Qiu & Zhu (2021), called $\pi$-PRL. Their algorithm uses TRPO (Schulman et al., 2015) to learn a tree policy in two phases. In the first phase, the algorithm alternates between learning the probability distribution over the possible trees ("architecture iteration") and learning the parameters of each tree ("program iteration"). A training step is a sequence of (1) gathering data from rollouts, (2) an architecture iteration, (3) gathering data from rollouts, (4) a fine tuning iteration. At the end of this first "fusion" phase, the most probable tree is extracted, and its parameters are fine-tuned during a second "fine-tuning" phase. We refer to the original paper for more precise details about the training scheme of their method. The authors propose several controllers to apply in the leaves of the tree. We focus on affine controllers as they are the one that compare to our method, since each of HC's subpolicies are affine functions.

We focus on the Cheetah run experiment from DM Control. We modify the code provided by the authors to be able to use affine policies in each leaf, similarly to HC. We parameterize the programmatic policy to consider the full state when choosing which branch to follow at an intersection (Eq. 1 of their paper). We repeat the experiments over 10 seeds.

We use the original paper's hyperparameters when available, and we detail the full list in Table 4.

We illustrate the results of the algorithm resulting in a soft tree and a hard (discrete) tree respectively in Fig. 21 and Fig. 22. In both cases, the algorithm does not manage to learn a strong policy in Cheetah run, despite having access to 15 times as many samples as HC and SAC (15M vs 1M, respectively). The algorithm converges in fact to a linear architecture, hence the same plots for the soft and hard trees.

We also plot the performance of a neural policy with two hidden layers and 256 hidden units each trained with TRPO on Cheetah run to showcase a likely upper bound on the performance of $\pi$-PRL, in Fig. 23. Note that we use 10 times more training samples for TRPO than for HC or SAC in order

---

[2]https://github.com/denisyarats/pytorch_sac

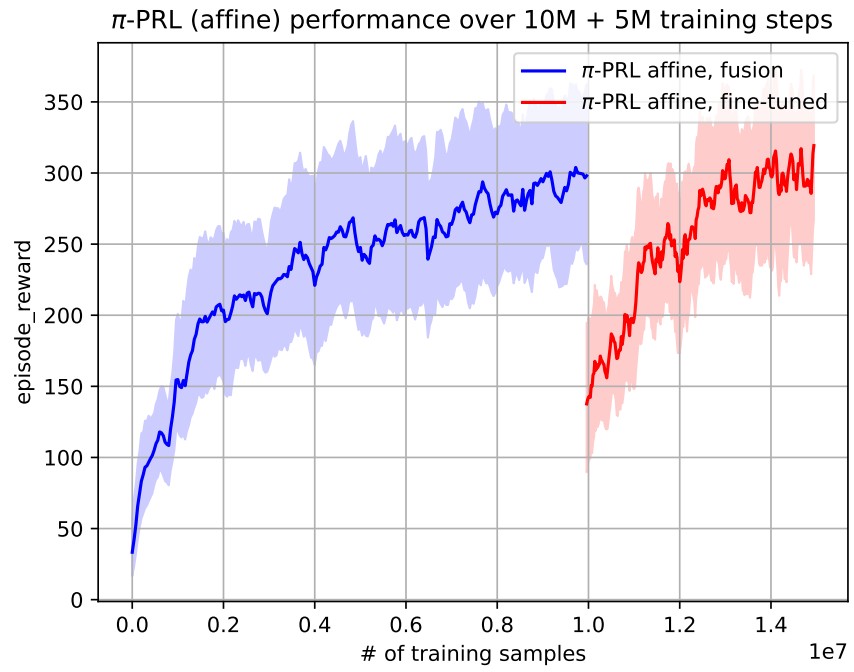

Figure 21: In Cheetah run, $\pi$-PRL (affine) did not manage to learn a strong policy.

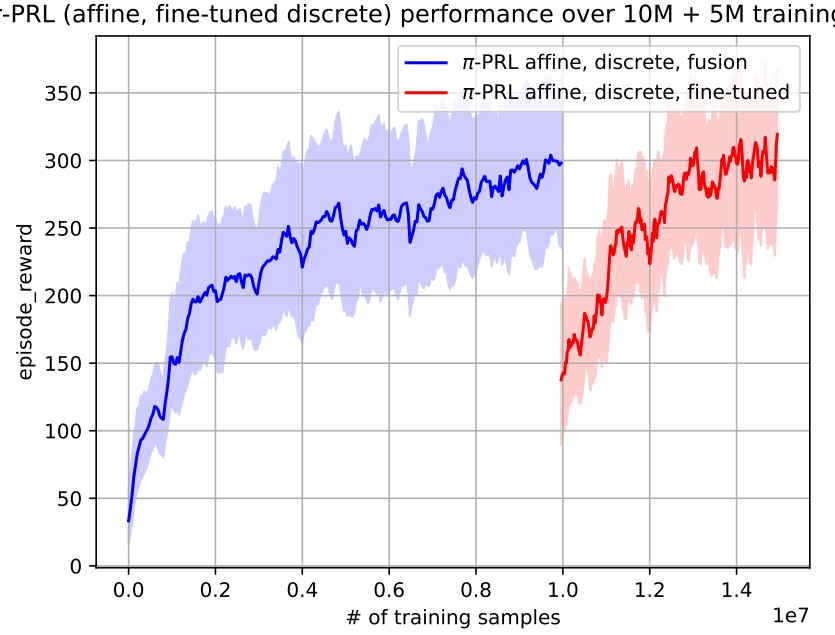

Figure 22: In Cheetah run, $\pi$-PRL (affine and hard decision tree) did not manage to learn a strong policy.

| Hyperparameter name | Value |
|---|---|
| Action repeat | 1 |
| Discount factor $\gamma$ | 0.99 |
| Max graph depth | 6 |
| GAE $\lambda$ | 0.97 |
| Num of consecutive architecture iterations | 1 |
| Num of consecutive program iterations | 1 |
| Total number of "fusion" training step | 1000 |
| Total number of "fine-tuning" steps | 1000 |
| Samples per data gathering rollout | 5000 |
| Baseline architecture | [64,64] |
| Baseline num epochs | 2 |
| Baseline learning rate | $1e-3$ |
| Baseline batch size | 64 |

Table 4: List of hyperparameters in the programmatic policies comparison.

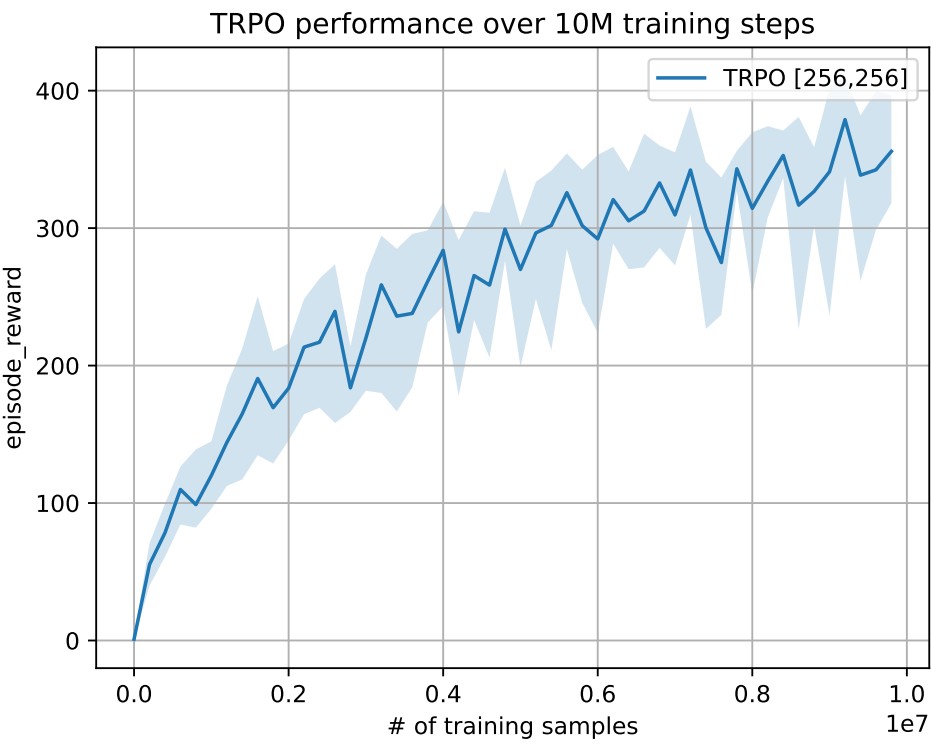

Figure 23: In our setting, TRPO plateaus slightly below 400.

to obtain a good performance with TRPO (10M vs 1M). TRPO and $\pi$-PRL plateau below an episode return of 400 (lower than SAC or HC).

Since the results of the algorithm are high variance (potentially because of the usage of policies of different capacity depending on the probability distribution over architectures that is learnt), we analyze in Fig. 24 the performance of the best performing seed in the affine, discrete case. We notice that the performance rises quickly to around 400, but stays stuck in this local optimum.

Finally, we note that $\pi$-PRL is restricted to small architectures (Orfanos & Lelis (2023)), while HC can use deep networks to partition the input space, and instead restricts the number of unique affine sub-policies that can be used.

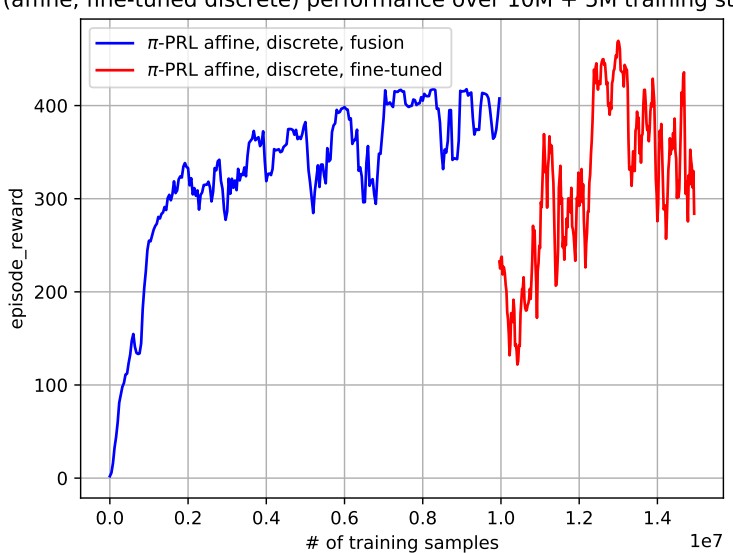

Figure 24: Performance of the best seed, which gets a maximum score of 450.

### D.8.1 ADAPTING PI-PRL TO SAC

We adapt $\pi$-PRL to be trained with the SAC algorithm. We re-use the hyperparameters from Table 3 and repeat the experiment 8 times. We used half of the training samples budget for the first fusion phase, and half for the second fine-tuning phase. We note that we test $\pi$-PRL (SAC version) unfairly compared to HC, since $\pi$-PRL is in this experiment a soft decision tree, as mentioned in Eq. 1 of Qiu & Zhu (2021). As a consequence, the predicted actions from $\pi$-PRL are a mixture of all the actions in each leaf, which impedes interpretability. On the other hand, HC uses a single linear sub-policy at each timestep.

We show the results in Fig. 25. We note that, besides the first 200k timesteps, the SAC version of $\pi$-PRL seems to perform below HC. The parameterization that we chose for HC is able to quickly improve the quality of the policy, comparatively to the other algorithms. We recall that even besides this control experiment, as mentioned before, the same architecture of HC is able to handle high level planning tasks, while $\pi$-PRL does not. In addition, the end-user has the possibility with HC to change its performance interpretability trade-off by increasing the number of sub-policies to use for a marginal cost by simply increasing the size of the Gumbel layer, while the same operation comes at a great computational cost for $\pi$-PRL (Qiu and Zhu, 2022, Section 3, Complexity; Orfanos et al, Section 2., paragraph 2).

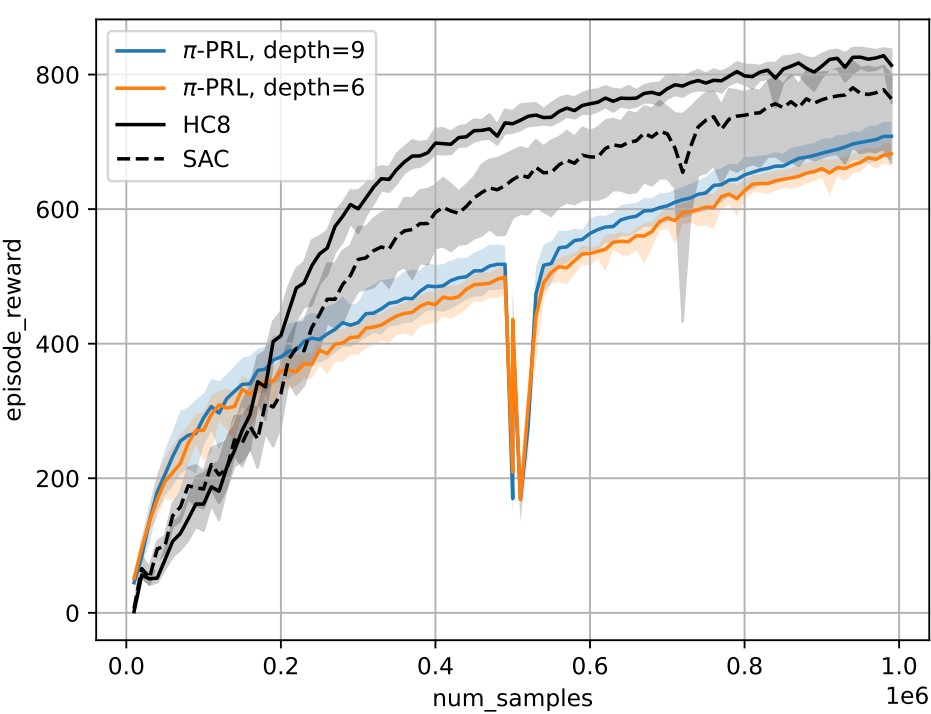

Figure 25: Comparison between the SAC version of $\pi$-PRL and HC.

# E  NAVIGATION EXPERIMENTS

## E.1  IMPLEMENTATION DETAILS

We base our experiments on the open-source code provided by RIS (Chane-Sane et al., 2021)[3], and use most of its default hyperparameters. We detail in Table 6 the common hyperparameters and the hyperparameters we change for the HyperCombinator.

**RIS algorithm**  RIS belongs to the family of goal-conditioned RL algorithms, that learn a policy $\pi(s, g)$. RIS aims at guiding the policy, during training, to produce the same action to reach a goal as the action to reach an imaginary subgoal located halfway through the trajectory. In this case, the notion of distance implied in "halfway" means that as many steps are needed to go from the current state to the imaginary subgoal than from the imaginary subgoal to the goal (as opposed to an Euclidean notion of distance that would not be suitable in mazes).

To do this guiding, a *prior policy* $\pi_{\text{prior}}$ is defined as an empirical moving average of the online, interacting policy. A neural network (denoted a *high-level policy* in the original paper) learns online to generate likely subgoals $s_g$ given a state $s$ and a goal $g$. Then, given state $s$, the policy $\pi(s, g)$ is constrained to stay close to the prior policy to reach subgoal $s_g$, *i.e.* $\pi_{\text{prior}}(s, s_g)$.

Similarly to SAC, the RIS actor is composed of an MLP on top of which two heads predicting the mean of the action predictive distribution, $\tilde{\pi}(x)$, and its log standard deviation $\log \sigma(x)$. $t'$ clamps $\log \sigma(x)$ between $\log \sigma_{\min}$ and $\log \sigma_{\max}$. The action is then sampled from a squashed normal, similarly to SAC. Two critics, each modeled by a MLP, are learnt with double Q learning. The hyperparameter $\alpha$ ((Chane-Sane et al., 2021, Eq. 9)) is fixed during training. The algorithm uses Hindsight Experience Replay (HER) (Andrychowicz et al., 2017) to facilitate training, and we refer to the RIS paper for further details on the specific implementation (Chane-Sane et al., 2021). All weight matrices are initialized using orthogonal initialization. All bias vectors are initialized to 0.

**HyperCombinator modifications**  We replace $\tilde{\pi}$ by a Gumbel network. Similarly to the control experiments, $\log \sigma$ is still defined by an MLP, but does not share parameters with $\tilde{\pi}$. As a consequence of the modelisation of $\tilde{\pi}$, the prior policy $\pi_{\text{prior}}$ also takes the form of a HC policy. We do not modify the rest of the algorithm, including the high-level policy that learns to predict the intermediate goals used for training. Due to memory constraints, we use a batch size of 1024 to train the HyperCombinator, instead of the base 2048. We train RIS with batch sizes of both 1024 and 2048 and ensure that the performance of RIS with 2048 (the base value) improves over the performance with a batch size of 1024. We detail in Alg. 3 the RIS algorithm and add in **blue** the modifications that we apply.

**Training details**  The agents interact with the environment for a maximum of 600 timesteps, after which the episode is interrupted.  We do not use Gumbel noise for any interaction with the environment, which guarantees that the most likely *sub-policy* is consistently selected. Therefore, sub-policies are stochastically chosen only during the update of the actor, in the `UpdateActorAndAlpha` routine. During training, the actions are sampled from the action predictive distribution $\mu(\mathcal{N}(\tilde{\pi}(s, g), \exp(t'(\log \sigma(s, g)))))$ to encourage exploration. We use three different types of regularization to prevent the Gumbel network from collapsing into the prediction of a single sub-policy:

- We increase the temperature of the Gumbel-Softmax, controlled by the "Gumbel temperature" parameter in Table 6.

- We regularize the entropy of the sub-policy assignations averaged over a batch, controlled by the $\lambda_{\text{assig}}$ hyperparameter.

- We penalize the magnitude of the last layer weights of the MLP preceding the Gumbel-Softmax, through the $\lambda_{\text{weight decay}}$ hyperparameter. This forces the input to the Gumbel-Softmax layer to have a smaller magnitude, and therefore encourages the selection of a more diverse set of sub-policies.

---

[3]https://github.com/elliotchanesane31/RIS

**Model choice**   We experiment with different combinations of the architecture of the Gumbel network (from [32,32] to [1024,1024,1024]), the sub-policy assignation entropy coefficient $\lambda_{\text{assig}}$ (between 0 and 0.1), the Gumbel temperature (from .5 to 10), the weight decay coefficient (between 0 and 0.001) and the RIS hyperparameter $\alpha$ (between 0.05 and 0.4) on the U-shaped maze and the $\omega$-shaped maze (we found that the relative simplicity of U-shaped maze compared to the other environments meant that design architectures useful to solve the U-shaped maze would not always generalize to the other tasks) and select the final set of hyperparameters based on the return curve. We then run the final experiment on all environments, keeping the same hyperparameters for all variants of the HyperCombinator (*i.e.* HC8, HC16 and HC64).

A notable difference with the control experiments is that we found that smaller Gumbel network architectures worked best. Therefore, we selected a [64, 64, 64] architecture for the navigation experiments, as opposed to [1024, 1024, 1024] for the control experiments. We have overall observed in both control and navigation experiments that a wider and deeper Gumbel network architecture usually requires a stronger regularization to perform well on the task. It is possible that increasing even more the regularization could help the [1024, 1024, 1024]-architecture solve the maze tasks. The success of the smaller architecture and the failure of the bigger one, which such a high regularization of the Gumbel network, is another clue that the key to solving the maze might be to find a trainable and sufficiently regularized architecture that efficiently combines the different sub-policies.

**Evaluation details**   During evaluation, all actors act deterministically using only the mean of the predicted action distribution $\mu(\tilde{\pi}(s, g))$, without exploration noise (as opposed to sampling from the predictive action distribution during training), which guarantees the piecewise-linearity of the policy. We remark that this departs from the base code, that evaluated stochastic agents.

Every 10000 steps, we roll out 5 evaluation episodes and report the mean success score, *i.e.* 1 if the agent reached the goal and 0 else. We report all results and curves using 10 seeds for each agent. We draw the mean performance as a colored line, as well as a 95% bias-corrected and accelerated bootstrap confidence interval in a lighter shade (9999 resamples).

**Compute**   We ran all the experiments on an internal cluster. All the GPUs were NVIDIA Tesla V100, with 16GB memory available. The CPUs were Intel(R) Xeon(R) CPU E5-2698 v4 @ 2.20GHz Each seed was allocated 1 GPU, 10 CPUs, and 64GB of RAM. We detail the compute budget to reproduce the experiments in Table. 5.

| Experiment | # models | # envs | # seeds | Avg. duration | Compute |
|---|---|---|---|---|---|
| Full results (Fig. 4) | 4 | 4 | 10 | 23 hours | 153 GPU days |
| Temperature ablation (Fig. 30) | 8 | 1 | 10 | 11 hours | 37 GPU days |

Table 5: Compute budget for the navigation experiments.

---

[4]Shared between the mean net and the log std net

[5]https://github.com/elliotchanesane31/RIS

[6]https://github.com/elliotchanesane31/RIS

---

**Algorithm 3** `RIS (with HyperCombinator actor)`

---

**Require:** Replay Buffer $\mathcal{D}$
**Require:** Actor parameters $\varphi$
**Require:** Double critic parameters $\eta_1, \eta_2$
**Require:** Double critic target parameters $\overline{\eta}_1, \overline{\eta}_2$
**Require:** High-level policy parameters
**Require:** Hyperparameters from Table 6
**Require:** $N$                                                      ▷ Maximum number of timesteps
**Require:** $s_0, g_0$                                                        ▷ Initial state
  **while** $t < N$ **do**
    $a_t \sim \mu(\tilde{\pi}(s_t, g_t))$
    $s_{t+1} \sim P(\cdot | s_t, a_t)$                           ▷ Sample the next state from the environment
    $r_{t+1} = R(s_t, a_t, s_{t+1})$
    $\mathcal{D} = \mathcal{D} \cup (s_t, a_t, r_{t+1}, s_{t+1}, d_{t+1})$     ▷ Update replay buffer; $d_{t+1}$ indicates a terminal
transition

    $(s, a, r, s', d, g) \sim \mathcal{D}$                          ▷ Sample from replay buffer using HER
    Launch routine `UpdateCritic`          ▷ See paper (Chane-Sane et al., 2021) and code[5]
    Launch routine `UpdateHighLevelPolicy`     ▷ See paper (Chane-Sane et al., 2021) and
code[6]
    Launch routine `UpdateActorAndAlpha` (see Alg. 4)
    $\overline{\eta}_1 = (1 - \text{critic ema ratio}) * \overline{\eta}_1 + \text{critic ema ratio} * \eta_1$
    $\overline{\eta}_2 = (1 - \text{critic ema ratio}) * \overline{\eta}_2 + \text{critic ema ratio} * \eta_2$
  **end while**

---

**Algorithm 4** `UpdateActorAndAlpha`

---

**Require:** $s, g$                          ▷ Batch of states and goals sampled from the replay buffer
  $a \sim \mu(\mathcal{N}\left(\tilde{\pi}(s, g), \exp(t'(\log \sigma(s, g)))\right))$     ▷ $\tilde{\pi}$ is a Gumbel network and $\log \sigma$ an independent
MLP instead of being a jointly parametrized MLP in the base case.
  $\mathcal{L}_{\text{assig}} = H(\text{mean}(\hat{a}(s, g)))$               ▷ Compute entropy of average sub-policy assignation
  $\mathcal{L}_{\text{weight decay}} = ||W||_2^2$          ▷ Magnitude of the Gumbel network MLP last layer weights $W$
  $Q = \min(Q_{\eta_1}(s, a), Q_{\eta_2}(s, a))$
  $\mathcal{L}_\pi = \text{mean}\left(\alpha \log \pi(a|s, g) - Q\right) - \lambda_{\text{assig}}\mathcal{L}_{\text{assig}} + \lambda_{\text{weight decay}}\mathcal{L}_{\text{weight decay}}$     ▷ Minimize the
magnitude of the Gumbel network last layer weights
  $\varphi = \varphi - \lambda_\pi \nabla_\varphi \mathcal{L}_\pi$

---

| Hyperparameter name | Value |
|---|---|
| *Common (RIS defaults)* | |
| Discount factor $\gamma$ | 0.99 |
| Success distance threshold | 0.5 |
| Burn in | 1e4 |
| Replay buffer size | 1e6 |
| Learning rate for the high-level policy $\lambda_{\text{high-level}}$ | 1e-4 |
| Learning rate for the critics $\lambda_Q$ | 1e-3 |
| Learning rate for the policy $\lambda_\pi$ | 1e-3 |
| Critic architecture | [256, 256] |
| Critic empirical moving average ratio | 0.005 |
| Critic target update frequency | 1 |
| High-level policy architecture | [256, 256] |
| sub-policy assignation entropy coefficient | 1e-4 |
| $\epsilon$ | 1e-16 |
| HER replay buffer goals ratio | 0.5 |
| $\lambda$ | 0.1 |
| $\log \sigma_{\min}$ | -20 |
| $\log \sigma_{\max}$ | 2 |
| *RIS actor-specific* | |
| Actor architecture[4] | [256, 256] |
| Batch size | 2048 |
| $\alpha$ | 0.1 |
| *HyperCombinator-specific* | |
| Gumbel net architecture | [64, 64, 64] |
| Batch size | 1024 |
| sub-policy assignation entropy coefficient $\lambda_{\text{assig}}$ | 0.01 |
| Weight decay coefficient $\lambda_{\text{weight decay}}$ | 0.0001 |
| Gumbel temperature | 7 |
| $\alpha$ | 0.3 |

Table 6: Full list of hyperparameters in the navigation experiments.

## E.2 ADDITIONAL POLICY VISUALIZATIONS

In Fig. 5 we illustrated a typical sequence of sub-policies followed by HC16 on the S-shaped maze. In this section, we propose a wider variety of sub-policy sequences, showing examples where the emergent temporal abstraction appears clearly, and some where this temporal abstraction is harder to spot, or, interestingly, is lost over training. For each situation, we show sub-policy sequences taken during evaluation after .5M, 1M, 1.5M and 2M training steps.

**HC8, U-shaped maze (Fig. 26)** This figure shows a well-performing HC8 policy in U-shaped maze. We note the clear separation of three phases in the bottom plot, as the quadruped learns to go down the corridor in the first phase, using mostly sub-policy 0 and 1, then turns right and navigates the bottom corridor with sub-policies 2 and 3, before re-using sub-policies 0 and 1 to go up the right corridor and solving the maze.

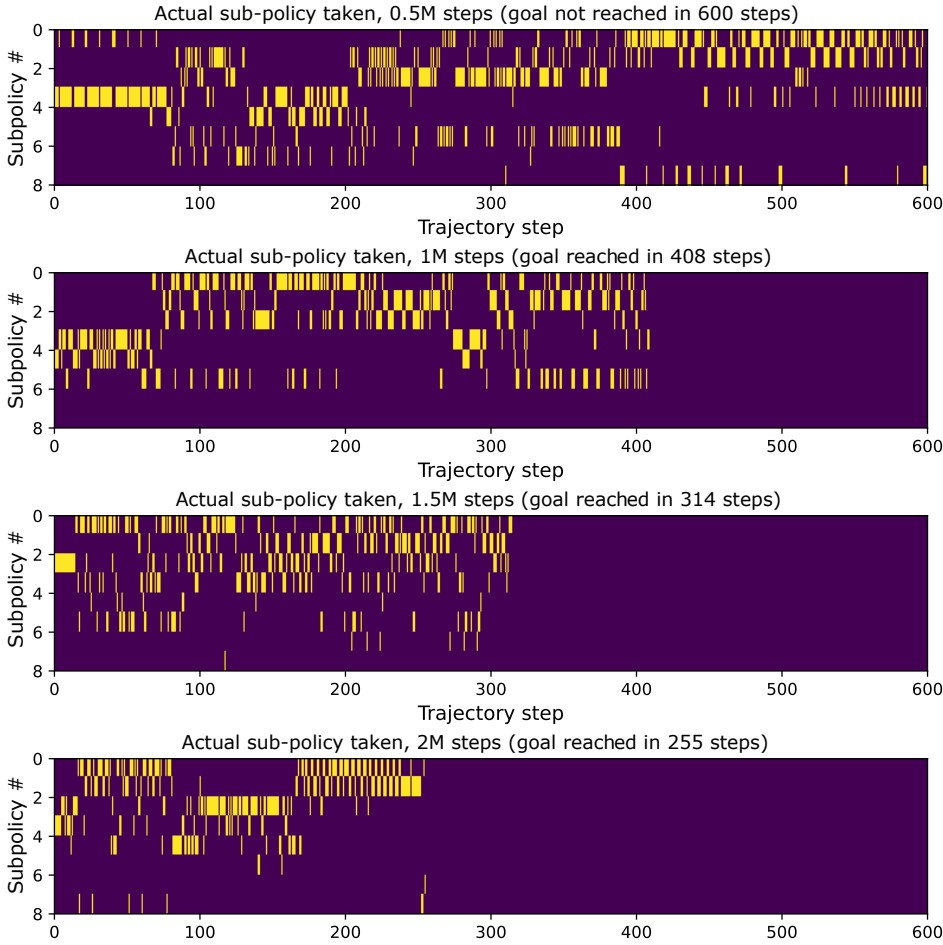

Figure 26: Example of a well-performing HC8 agent in the U-shaped maze.

**HC64, $\omega$-shaped maze (Fig. 27)** This figure illustrates the policies learnt by HC64 on the challenging $\omega$-shaped maze. We particularly note the appearance of "diagonals", where the same sequence of sub-policy is repeated several times. These diagonals recall the sub-policy sequence obtained for Cheetah run in the control experiments, as the agent learnt to chain its sub-policies to move itself. After 100 timesteps in the bottom figure (final evaluation after 2M training steps), HC64 switches from one set of repeated diagonals to another, and barely re-uses the sub-policies forming the first set of repeated diagonals until the end of the trajectory. We conclude that the first set of repeated diagonals was specialized to the first phase of the trajectory.

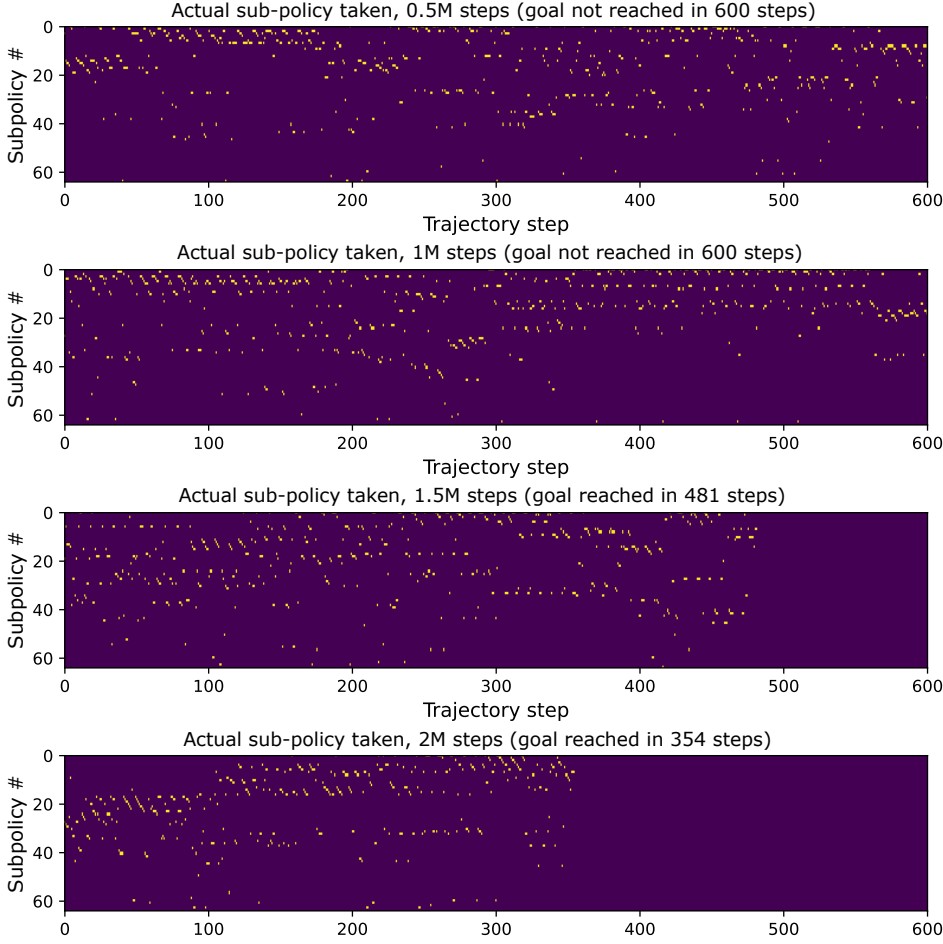

Figure 27: Example run of HC64 solving the $\omega$-shaped maze. We note the appearance of "diagonals" that are reminiscent of the control experiments.

**HC64, U-shaped maze (Fig. 28)**    We have seen in the previous example that a structure emerged in the sub-policies, learning to repeat chains of sub-policies ("diagonals") to move itself. Fig. 28 illustrates the possible effects of long training on this structure. This structure is particularly present after .5M timesteps, and the sub-policies in the second part of the trajectory are not used during the first part of the trajectory. However, as training progresses, the structure is progressively forgotten, which makes the bottom sub-policy sequence plot harder to read. One possible explanation is that the agent learns to "overfit" the task, resulting in a lower re-use of sub-policies, but a better performance (as evidenced by the progressively lower number of timesteps required to solve the task). This might also be due to the high regularization that we enforced on the Gumbel network, forcing agents to use as many sub-policies as possible.

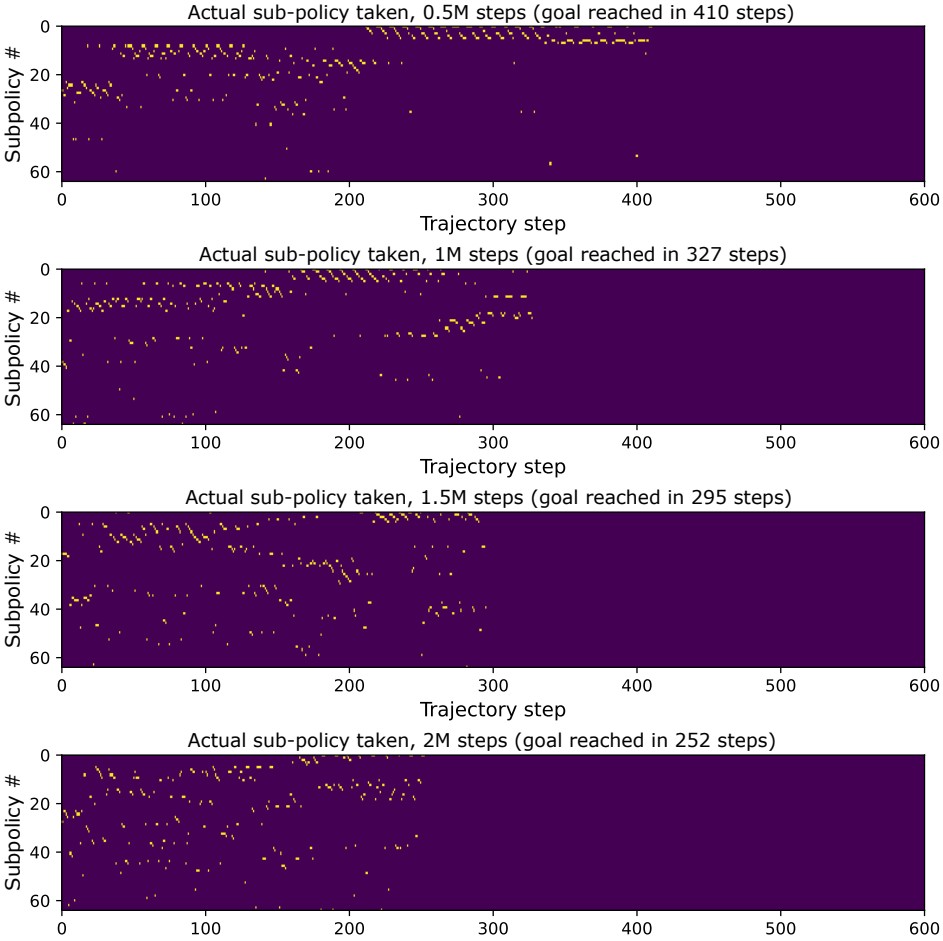

Figure 28: The learnt structure (top plot) can get lost as the agent learns to solve efficiently the maze (bottom plot).

**HC16, Π-shaped maze (Fig. 29)**    Fig. 29 finally provides an example of a policy where structure is difficult to observe. HC16 learns to solve the Π-shaped maze after 1.5M timesteps, but no particular structure emerges. In addition, we see one example of HC16 forgetting how to solve the maze in the bottom plot. We note that despite this fact, the HyperCombinator architecture guarantees that we can bring transparency to the interaction of the agent with the environment, conditioned on the knowledge of the sub-policy chosen. Notably, we still have access to all the sub-policies that define how HC16 interacts with the environment.

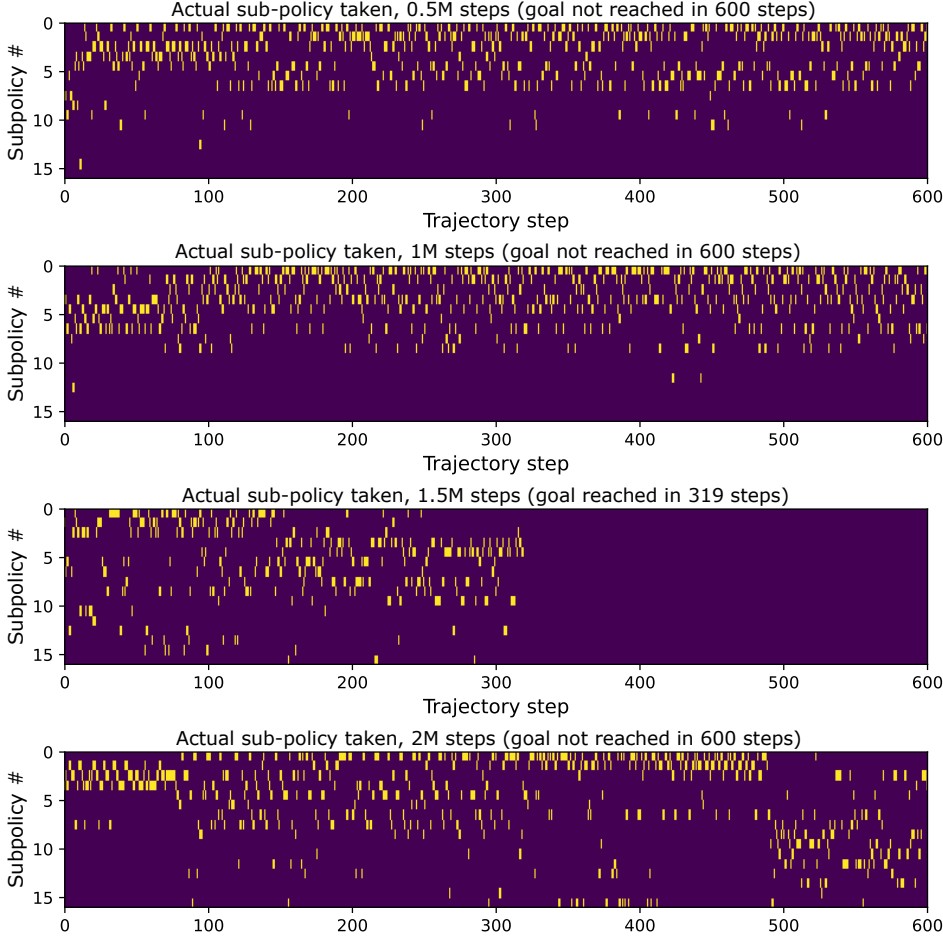

Figure 29: HC16 can learn to solve the maze without displaying obvious structure in its sequence of sub-policies (3rd plot).

### E.3 EFFECT OF GUMBEL TEMPERATURE ON PERFORMANCE

We run an ablation to see the effect of changing the temperature hyperparameter, all other hyperparameters being held the same as in the navigation part of Sec. 4. We evaluate the HyperCombinator variants for temperature values of .5, .66, 1, 2, 3, 5, 7 and 10 in the U-shaped maze. In general, a higher temperature value tends to increase the stochasticity of the Gumbel network, meaning that the Gumbel network will be more uncertain of which sub-policy to choose (and will also increase gradient sharing during the backward pass).

We visualize the results of the ablation in Fig. 30. Overall, low temperature values impede learning, and the resulting agents do not succeed in the U-maze. Higher temperature values led to better performing agents, though the overall return curve remains noisy in all cases.

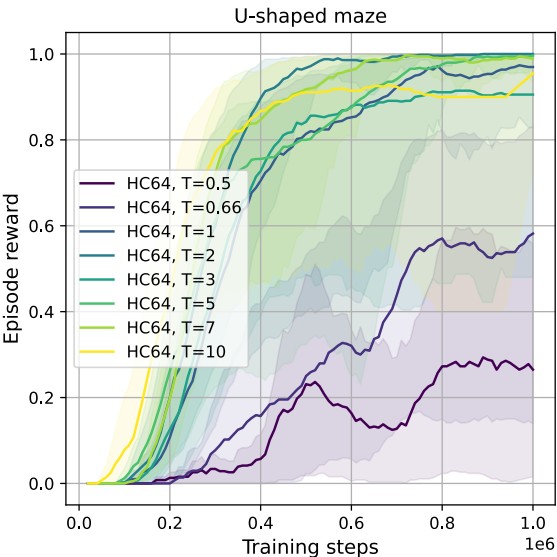

Figure 30: Low temperature values lead to insufficient regularization and HC64 fails to solve the maze. Increasing the temperature overall leads to better performance, except for the highest temperature.

