# OpenReview forum: "Piecewise Linear Parametrization of Policies: Towards Interpretable Deep Reinforcement Learning"
_ICLR.cc/2024/Conference — ICLR 2024 poster_

### Official Review · Reviewer_j8Wd · 2023-10-30

**Soundness:** 2 fair
**Presentation:** 3 good
**Contribution:** 2 fair
**Rating:** 6
**Confidence:** 5

**Summary:**

The paper presents a novel architecture for actor-critic algorithms with the goal of increasing the interpretability of the learned policies. The proposed architecture learns a model that maps each input to a linear function, which is used to predict the agent's action. The underlying assumption is that, if we can understand the set of linear functions used to produce actions, then the model is "partially" interpretable.

The new architecture, HyperCombinator (HC), is evaluated with SAC, where the actor model is replaced with an HC. The SAC-HC algorithm is evaluated on the DeepMind Control Suite benchmark. The empirical results show that the performance of SAC-HC in terms of sample efficiency is similar to that of SAC.

The paper also presents plots showing how different linear functions are used across different episodes of problems such as Cheetah.

**Strengths:**

Policy interpretability is an important topic, since policies we can understand and verify are important in real-world scenarios.

It is also interesting to see empirical evidence that it is possible to perform well in commonly used benchmarks with models with less capacity.

Another strength of the paper is its clarity and ease of understanding. While the topics discussed aren't necessarily simple, the authors have done a good job presenting their results.

**Weaknesses:**

I have two main concerns with this paper. The first is about a whole body of work on programmatic policies that the paper overlooks. The second is about the weak evaluation related to the interpretability of the proposed model. I'll detail each concern below.

**Missing Related Work**

Below are some related works that I think the paper missed. The paper should cite some of these to better place its contributions within the existing literature. Others should serve as baselines in the experiments, as I explain later.

There's a line of research called Programmatically Interpretable RL (PiRL), see Verma et al. The goal in this line of research is to create programs that encode policies for RL problems. One of the key motivations behind PiRL is interpretability. Both Verma et al. and Bastani et al. use imitation learning to train interpretable models. Later, Qiu and Zhu found a way to learn similar policies with a fully differentiable approach, so there's no need for imitation learning. The architecture of Qiu and Zhu is essentially an oblique decision tree, which could also be trained with ReLU neural networks, as discussed in the work of Lee and Jaakkola and of Orfanos and Lelis.

The properties of ReLU networks share a lot in common with what's mentioned in the paper for the HC architecture:

1. "We make the functions $a$ and $\theta$ explicit through a new parametrization of the NN."

This is also possible with ReLU networks. In these networks, the function $a$ is the set of weights in the hidden layers, while $\theta$ is the function the model learns when the activation pattern is fixed (see Lee and Jaakkola for more). So, what makes the HC different from ReLU networks?

2. "We explicitly control the number of unique sub-policies of $\pi$ through the dimension $dG$ of the Gumbel-Softmax layer."

This is also the case with the number of neurons in ReLU networks. Another way to reduce the number of unique sub-policies in ReLU networks is to use a single hidden layer with strong L1 regularization (refer to Orfanos and Lelis).

3. "Policies modeled with the HyperCombinator differ from MLPs in that they usually aren't continuous at the border between linear regions."

I'm not sure why this is crucial, but ReLU networks also switch the linear function that gives the prediction when you move from one region to another.

4. "Our approach is locally interpretable, explaining the sub-policy applied to any given example."

This is true for ReLU networks mapped to Oblique Decision Trees too. The function that gives the prediction is at the leaf node. The path in the tree might not be easy to understand, but the linear function at the leaves is just as clear as those in the HC architecture.

Given all these similarities, small ReLU networks should be used as baselines in the experiments. Specifically, the architecture by Qiu and Zhu showed strong results on the same benchmark problems used in this paper.

Other papers I mention below might be less crucial but are related. So, it would be good to see where this paper stands within the broader literature. For example, Inala et al. discuss how to learn interpretable finite state machines for RL problems. Koul et al. learn FSM policies for RL problems from recurrent networks. Aleixo and Lelis look at programmatic policies in multi-agent RL. Trivedi et al. and Liu et al. learn a latent space of a domain-specific language which can be used to search for programmatic policies. All these papers explore potentially interpretable policies for RL problems, making them relevant to this submission.

**Lack of Evaluation on Interpretability**

The paper mostly gives anecdotal evidence when it comes to the interpretability of the policies. I appreciate the plots showing how the sub-policies work, but they're just examples. For some reason, the literature on interpretable policies doesn't focus much on evaluating interpretability. The papers I've listed below are weak in this area too. But none of them emphasize interpretability as much as this one does. The big unanswered question is: are these policies really interpretable, and if so, to whom?

**References**

1. Osbert Bastani, Yewen Pu, and Armando Solar-Lezama. Verifiable reinforcement learning via policy extraction. In Proceedings of the International Conference on Neural Information Processing Systems, pages 2499–2509. Curran Associates Inc., 2018.

2. Jeevana Priya Inala, Osbert Bastani, Zenna Tavares, and Armando Solar-Lezama. Synthesizing programmatic policies that inductively generalize. In International Conference on Learning Representations, 2020.

3. Anurag Koul, Alan Fern, and Sam Greydanus. Learning finite state representations of recurrent policy networks. In International Conference on Learning Representations, 2019.

4. Guang-He Lee and Tommi S. Jaakkola. Oblique decision trees from derivatives of relu networks. In International Conference on Learning Representations, 2020.

5. David S. Aleixo and Levi H. S. Lelis. Show me the way! Bilevel search for synthesizing programmatic strategies. In Proceedings of the AAAI Conference on Artificial Intelligence. 2023.

6. S. Orfanos and Levi H. S. Lelis. Synthesizing programmatic policies with actor-critic algorithms and relu networks, 2023.

7. Abhinav Verma, Vijayaraghavan Murali, Rishabh Singh, Pushmeet Kohli, and Swarat Chaudhuri. Programmatically interpretable reinforcement learning. 2018.

8. Abhinav Verma, Hoang M. Le, Yisong Yue, and Swarat Chaudhuri. Imitation-projected programmatic reinforcement learning. In Proceedings of the International Conference on Neural Information Processing Systems. Curran Associates Inc., 2019.

9. Wenjie Qiu and He Zhu. Programmatic reinforcement learning without oracles. In International Conference on Learning Representations, 2022.

10. Dweep Trivedi, Jesse Zhang, Shao-Hua Sun, and Joseph J Lim. Learning to synthesize programs as interpretable and generalizable policies. Advances in neural information processing systems, 34:25146–25163, 2021.

11. Guan-Ting Liu, En-Pei Hu, Pu-Jen Cheng, Hung-Yi Lee, and Shao-Hua Sun. Hierarchical programmatic reinforcement learning via learning to compose programs. arXiv preprint arXiv:2301.12950, 2023.

**Questions:**

1. How does HC architecture compare with other works from the literature, especially those on the PiRL line of work?

2. How can one properly evaluate the interpretability of these models?

---

> ### Author Response · Authors · 2023-11-15
>
> We thank the reviewer for their detailed review of our submission and their precise feedback.
> We address their concerns in detail below.
>
>
>
> ### Relationship of HC with ReLU networks
> We agree that HC is indeed extremely linked with MLP-ReLU networks (abbreviated as ReLU networks in the following), as the latter is actually the starting point from our architecture (first paragraph of Sec 3.1). However, we made some strong changes to the ReLU network architecture in order to achieve desirable interpretability properties which we identified at the end of Sec. 2 and beginning of Sec. 3.
> Overall, the differences between ReLU networks and HC are best exemplified by Fig. 1 (right):
> ReLU networks induce (1) a complex partition of the input space and (2) a different linear sub-function in each linear region (one color per linear sub-function).
> HC also induces (1) a complex partition (useful to solve hard tasks from scratch), but (2) only exhibits a small number of linear sub-functions (illustrated by using only 4 colors, hence 4 linear sub-functions). This is in stark contrast with ReLU networks of all but the smallest architecture.
>
> In addition, we have actually compared ReLU networks (including with small architectures) with HC in our submission:
> In Appendix B, Fig. 6, we show evidence that HC reaches a better performance-interpretability trade-off than ReLU networks for a wide range of architectures. This leads HC to a high performance while using a very small number of linear sub-policies.
> In Appendix C, Fig. 7, we show how a ReLU network uses essentially a new sub-policy at each timestep, on the contrary to HC.
>
> We now answer your points detailing the differences between HC and ReLU networks, following your points:
>
> ### 1. Explicit formulation of $a$ and $\theta$
>
> We agree that it is possible to recover $a$ and $\theta$ for ReLU networks, as we indicated in the beginning of Sec 3.1. However, these functions do not appear explicitly in the computation of the output $f(x)$, and need to be recovered (in the case of the local linear coefficients, by taking the input Jacobian of $f$).
> On the contrary, we make $a$ and $\theta$ explicit within the computation made by the neural net, by composing a Gumbel network $a$ and the local linear coefficients $\theta$. The explicit modeling of $a$ within the forward pass of the actor lets us set the *exact number* of sub-policies as a hyperparameter, unlike ReLU networks (we expand on this last statement in the answer to your second point).
>
> Following your comment, we modified the manuscript (Sec 3.1) to clarify that $a$ and $\theta$ now appear explicitly in the computation (in purple).

---

> > ### Author Response · Authors · 2023-11-15
> >
> > ### 2. Controlling the number of sub-policies
> > It is possible to upper bound the number of sub-policies of ReLU networks through their architecture. However, the exact number of sub-policies depends on the activation patterns that are learnt (a network with only parameters with value 0 has only one linear region). Therefore, there is no hyperparameter setting directly how many linear sub-policies the network can use in ReLU networks, on the contrary to HC, where we can set this hyperparameter (the output size of the Gumbel network, $d_G$).
> >
> > Another issue with ReLU networks is that the number of linear regions can grow very quickly with the width and depth of the network. While it is only a theoretical upper-bound, this is illustrated in (Montufar, 2014). Some more recent work has shown evidence that the actual number of linear regions used by the network grows less quickly (Hanin & Rolnick, 2019). Therefore, we performed in Appendix B an extensive empirical study to evaluate, in the DM control benchmark, how many unique sub-policies a ReLU network actually uses. We did this for a wide range of depth and width of ReLU networks.
> >
> > The results are illustrated in a Pareto plot, in Fig. 6. We analyze in detail these results in Appendix B, both theoretically (B.1) and empirically (B.2). In summary, HC obtains a great performance with far fewer sub-policies used, because we are able to constrain this number explicitly with $d_G$. In comparison, ReLU networks use far more sub-policies to reach a comparable performance.
> > We also show in Fig. 7 (Appendix C) the cumulative number of unique sub-policies re-used over a trajectory, which shows that ReLU networks use essentially a new linear sub-policy at almost every timestep (950/1000).
> >
> > Finally, we note that while one could add an L1 penalty on the parameters of the neural network to hopefully limit the number of linear regions used, the exact number of sub-policies expressed by the network would not be known a priori. On the contrary, HC lets us set before training the strict number of linear sub-policies that we are willing to allocate to the problem.
> > Still, following your advice, we looked more carefully if adding an L1 penalty enabled a higher performance while keeping the number of linear sub-policies used in a trajectory low, extending the analysis of Appendix B.
> > We added the results in a new Appendix B.3 section, which we briefly comment now.
> > On our benchmark, we did not find that adding an L1 penalty significantly improved the performance/interpretability trade-off. We tried the value mentioned in the Orfanos paper, as well as more general ones. The penalized policies were either using a very small number of linear sub-policies for a poor performance, or a high number for a high performance. In comparison, HC uses few linear sub-policies while reaching a high performance.
> >
> > ### 3. Continuity
> > Thank you for your comment. We mean to say that HC usually presents a discontinuity at the border between linear regions, while ReLU networks do not because they express continuous functions (and are therefore continuous at the border between linear regions, even though they are indeed usually not differentiable there). HC is in general neither continuous nor differentiable at the border between linear regions.
> >
> > We included this difference in the class of functions represented for exhaustivity, and did not intend to present this as an advantage of HC over ReLU networks. We verified that the non-continuity of HC was not an issue for the stability of the policy (Appendix D.6).
> >
> > ### 4. Local interpretability
> > Thank you for your remark. We agree that shallow (oblique) decision trees are also locally interpretable. However, deeper trees tend to weaken this property, as each leaf receives less examples, meaning that the coefficients of the linear function that is applied to a leaf are only applied to a few examples. Therefore, to ensure that the linear coefficients apply to a sufficient number of inputs, one solution is to regularize the depth of the tree.
> >
> > Similarly, all but the smallest ReLU networks lead in general to a high number of sub-policies used in practice (Appendix B and C), which significantly weakens their local interpretability.
> > On the other hand, we achieve with HC a similar effect to shallow decision trees by explicitly limiting the number of linear sub-policies that can be applied to a given example to only a small number. This forces the linear coefficients to be re-used for several inputs, essentially ensuring that the explanations provided by the coefficients are valid for a wide number of inputs.

---

> > > ### Author Response · Authors · 2023-11-15
> > >
> > > ### Relationship to programmatic policies
> > > We thank the reviewer for the references that they provided us.
> > >
> > > We agree that Programmatically interpretable RL is an area relevant to our research, and we updated our manuscript to include the references in our related works. We note that there are several significant differences with our approach. As you mention, both the papers from Bastani et al. and Verma et al. use imitation learning, while we directly train an interpretable policy by interaction, from scratch, setting our work within functional interpretability rather than distillation (we quickly touched on the distinction in Appendix A).
> > >
> > > The article from Qiu and Zhu is very interesting and closer to our work. A major difference with our work is that in most of their control experiments from Section 3 and 4, Qiu and Zhu’s algorithm assumes access to pre-trained abstract policies performing abstract actions such as going straight or turning (which they call ensemble policies). Their algorithm then learns to combine them for high level planning. They do mention affine policies in Section 2., but show that they perform much worse in Table 2. On the contrary, HC learns jointly the affine sub-policies (low-level control) and the high-level planning, from scratch. HC manages to solve complex tasks despite needing to discover by itself what movements are useful to achieve its goal. Moreover, as you mention, their policy is ultimately learned as an oblique decision tree where the leaves execute pre-trained abstract actions (whereas ours could be assimilated to an oblique decision tree where the leaves execute linear sub-policies), but as far as we understand, they do not use a neural network to model this tree (which could lead to additional constraints, such as having to keep the actor small (Orfanos 2023)).
> > >
> > > The work from Orfanos and Lelis was submitted to arXiv past the “contemporaneous” deadline of ICLR, but we are happy to discuss it now and include it in the literature review. Thank you for mentioning it to us. The authors do use a neural network actor in an actor critic setting, similarly to us. They then translate it into a programmable policy expressed as an oblique decision tree. They however have to restrict themselves to small actor architectures to prevent their tree from becoming uninterpretable, as mentioned in their introduction or experiments section. We are not bound by such a restriction, since we directly restrict the number of sub-policies that HC can exhibit while maintaining the complex partition induced by large neural networks. Moreover, we evaluate HC in both control and navigation settings, on the contrary to this paper.
> > >
> > > Overall, PiRL algorithms focus on the structure of the policies, expressing them for instance as finite state machines or small decision trees. In the latter case, the number of possible different behaviors of the policy (one per leaf) are implicitly defined by the height of the tree. The leaves apply abstract, discrete actions (Trivedi 2021, Qiu & Zhu 2022, Liu & Hu 2023) in navigation settings, or linear / PID sub-policies in control settings. In comparison, HC learns a small, pre-determined fixed number of linear sub-policies, while keeping the structure of the policy highly flexible. As a consequence, HC learns to perform both low-level control and high-level planning at the same time.
> > >
> > > ### About the interpretability of our method
> > > We thank the reviewer for their question. As a main advantage, HC brings transparency to the decision process through the two levels of interpretability: the linear coefficients of the very few sub-policies, and the sequence of chosen sub-policies over a trajectory.
> > >
> > > This transparency is a tool that lets stakeholders get insights about what their policy is doing, and how it is interacting with the environment. We argue the added transparency improves the interpretability over regular ReLU networks, which model far more linear sub-policies (Appendix B and C). At the same time, HC’s performance does not collapse for very difficult mazes, provided that some interpretability is traded-off for performance by increasing the number of linear sub-policies.
> > >
> > > For instance, we think the transparency of HC gives domain experts (who have knowledge of the environment) a useful tool to understand the policy they work with. We illustrate this aspect with a small analysis in Appendix D.7. We briefly analyze the meaning of the coefficients of the linear sub-policies and show how to interpret them in light of the sequence of sub-policies chosen.
> > >
> > > Finally, we agree with the reviewer that examples remain subjective since the interpretability depends on the stakeholder. Simultaneously, we designed HC such that the stakeholder could decide for themselves the level of simplicity of the policy, by setting the number of sub-policies $d_G$.
> > > Therefore, we think HC is a valuable step in the direction of inherently interpretable policies that solve complex tasks.

---

> ### Comment · Reviewer_j8Wd · 2023-11-17
>
> Thank you for your reply and all the edits to the paper.
>
> The related work section has improved, which is nice to see. The comparison with ReLU networks in the Appendix B (Figure 6) is also in the right direction.
>
> The issue I still have with the paper is the lack of baselines. For example, Qiu and Zhu's work is very related and it could be used even with no skills in the leaves. Qiu and Zhu used these skills in their experiments, but they are not needed to run their method. I am sorry for suggesting yet another baseline after the first round of reviews, but it just occurred to me that network distillation could also be a good baseline. You start with a large network with many unique regions and distill it into a much smaller network with fewer regions. A comparison with these methods would be informative.

---

> > ### Author Response · Authors · 2023-11-20
> > **2 new comparisons following your suggestions**
> >
> > Dear reviewer,
> >
> > Thank you for your answer and for your suggestions, which we followed over the past few days.
> >
> > Overall, we performed the two proposed experiments comparing (1) HC to the $\pi$-PRL policy from Qiu and Zhu and (2) HC to a distilled small actor. We found that both methods did not lead to a successful policy in Cheetah run, on the contrary to HC. We have updated the appendix with two detailed new sections to document our results in the comparison with programmatic (Appendix D.8, Fig. 21+22) and distillation (Appendix B.4, Fig. 7) policies. We now summarize our findings.
> >
> > ### Comparison to programmatic policies
> > Following your suggestion, we compared HyperCombinator with the $\pi$-PRL policy from Qiu and Zhu (2022).
> > We dedicated quite some time to understand and modify the code they provided online, which uses TRPO as the algorithm to train their policy, to be compatible with affine leaves and the Deepmind control suite. We applied their technique to the Cheetah run task, where we used affine controllers in each leaf of the tree. We tested two versions of the algorithm, one producing soft decision trees (using Eq. 1 of their paper) and one using hard decisions at each branching, in order to obtain a hard, regular decision tree (reflecting Fig. 3 of their paper). We also trained a TRPO algorithm with a neural policy (two layers of 256 units) in order to provide a comparable baseline to $\pi$-PRL.
> > We illustrate the results in Fig. 21 and 22 and comment on them, along with the experiment details, in Appendix. D.8. We also analyzed the behavior of the best performing seed, given that the results had high variance.
> >
> > In summary, we found that $\pi$-PRL did not match the performance of the baseline TRPO, which itself performs well below HC in this setting.
> >
> > ### Comparison to distilled policies
> > In the second experiment, we compared as suggested the HyperCombinator with a student policy that was trained to imitate an expert teacher policy.
> > We trained the student (a one-hidden layer network with 32 hidden units, as the network of Orfanas et al (2023)) to match the predictions of the teacher (a SAC agent with two hidden layers of 1024 units trained to convergence) by minimizing the mean squared error between their predicted actions. We used as dataset the states encountered by the teacher during its training run. Each state in the dataset is then labeled with the prediction of the fully trained teacher. We tried a variety of batch sizes and considered regularizing the weights of the teacher, similarly to Orfanas et al (2023).
> >
> > We illustrate the results in Fig. 7 of the manuscript. Despite a successful minimization of the mean squared error (Fig. 8, log scale), the students fail to obtain a satisfactory performance at Cheetah run.
> > We think the main reason for this catastrophic failure of the students is due to a mismatch between the objective they are trained towards (minimization of the action prediction error between the teacher and the student) and the purpose of the policies (maximize the return in Cheetah run). Because of their considerably lower capacity, students necessarily make approximations in order to fit the teacher. Since we cannot control in which states these approximation errors are made, it is possible for a student to poorly approximate a key state, such that the student is thrown off its trajectory when rolled out in the environment, even though it obtained a good mean squared error overall.
> > We think this issue is a key problem of distillation, as corroborated by Qiu and Zhu (2022)  (Section 6). We explored in Fig. 9 a possible fix, which is to include inputs seen by the student in the training data (Verma, 2018). This did not improve the performance of the student.
> >
> > On the other hand, the HyperCombinator is trained with reinforcement learning directly on the task. Therefore, it does not suffer from this mismatch and directly optimizes the desired objective (the cumulative sum of rewards), and as a consequence manages to learn a small policy that performs very well.
> >
> >  _
> >
> > We hope that these two additional comparisons will convince you of the value of our HyperCombinator approach in learning actors that interact with very few sub-policies with the environment, while reaching high performance.

---

> > > ### Comment · Reviewer_j8Wd · 2023-11-20
> > >
> > > Thank you for the effort to provide extra empirical results.
> > >
> > > I am not sure the comparison between $\pi$-PRL is fair because it uses TRPO, while HC uses SAC. Qiu and Zhu also used SAC for some of their experiments (see Table 3 of Appendix C of their paper). I understand you tried to compensate for this by providing TRPO more training samples. Do you think this is still a fair experiment to run?
> > >
> > > Regarding the imitation learning results, the common trick is to use DAgger as the learning procedure. I believe it is what you called "augmentation". Please see [1] for more information.
> > >
> > > [1] Ross, S., Gordon, G. J., & Bagnell, D. (2011). A Reduction of Imitation Learning and Structured Prediction to No-Regret Online Learning. Proceedings of the Fourteenth International Conference on Artificial Intelligence and Statistics (AISTATS) 2011, Fort Lauderdale, FL, USA.
> > >
> > > Did you implement DAgger or something else?
> > >
> > > Another trick that goes into the imitation learning process (used in Verma et al.'s work) is to use a loss function that mixes both the imitation part and the reward part. You want to use the imitation signal to learn faster but without losing track of the reward signal.

---

> > > > ### Author Response · Authors · 2023-11-21
> > > >
> > > > Thank you for your answer.
> > > >
> > > > ### Distillation experiment
> > > > We agree that there are a variety of other methods to incorporate expert knowledge to the training of a small policy (as we had mentioned in the last paragraph of Appendix B.4). These all involve specific algorithms with various inductive biases in order to tackle the imitation learning problem.
> > > > Crucially, all of these methods add the fundamental assumption of access to expert data or an expert policy, while the HyperCombinator does not. The HyperCombinator is instead learnt from scratch, without any privileged information. Therefore, we do not think that it would be fair to compare the HyperCombinator to the imitation learning literature.
> > > >
> > > > We do not claim that the HyperCombinator outperforms in general imitation learning methods, as these two methods make significantly different assumptions; we simply added the experiment of Appendix D.4 following your request, as we agreed it was an interesting addition, to see how the HyperCombinator fares against a simple distillation baseline.
> > > >
> > > > Regarding input augmentation: we called input augmentation the fact of adding training examples generated from the student policy (with expert labels from the teacher) to the regular training examples generated by the teacher policy (also labelled with the expert predictions). This is related to DAgger in that it involves both the student and the teacher in the generation of the trajectories, but it alternates between student trajectories and teacher trajectories instead of mixing the corresponding policies.
> > > >
> > > > ### Programmatic experiment
> > > > To be clear, the point of our comparison in Appendix D.8 was to show that $\pi$-PRL (TRPO version) was losing performance compared to the neural version of TRPO in DM Control’s Cheetah run task. This fact was already illustrated in Table 3 of Qiu & Zhu’s paper on the related MuJoCo HalfCheetah task. In turn, TRPO performs worse than SAC (and HC) in Cheetah run.
> > > >
> > > > In our understanding, $\pi$-PRL was only implemented in conjunction with the TRPO algorithm. In Appendix C of their paper, the authors compare their method to a neural version of SAC as a baseline in Table 3, but not that they used SAC to learn their own policy.
> > > > Therefore, while we agree that there are many elements separate pi-PRL (TRPO version presented in the paper) from HC, we felt it was reasonable to (1) confirm the performance difference between $\pi$-PRL (TRPO version) and neural TRPO on DM Control’s Cheetah run, and (2) observe how TRPO compared with SAC on this task. The additional data was provided in order to see if this could improve the performance of TRPO.
> > > >
> > > > Before making our last point, we would like to recall the two main divergences between $\pi$-PRL and HC:
> > > > - The affine version of $\pi$-PRL was proven to work on the control tasks from their Table 3 (Appendix C) only, and to fail on tasks requiring high-level planning (Section 5, Table 2 of their paper). On the contrary, we showed that HC with affine sub-policies performed particularly well on control tasks in our Section 4.1, and reasonably well in high-level planning navigation tasks (Section 4.2).
> > > > - The complexity of the inference grows quickly with the number of affine sub-policies (Section 3 of their paper, Complexity; Orfanos et al, Section 2., paragraph 2).

---

> > > > > ### Author Response · Authors · 2023-11-21
> > > > >
> > > > > Following your last request, we tried to adapt in a final experiment the existing code from Qiu and Zhu to be compatible with SAC, since the code provided online was presented only for TRPO.
> > > > > We insist that this is a novel result, since the original paper did not, as far as we understand it, present results with their policy trained with SAC. Crucially, the set up of his experiment is also unfair towards HC, because HC only selects one affine sub-policy at each timestep, while we evaluate in the experiment the soft-decision trees induced by $\pi$-PRL, which therefore realize a mixture of affine sub-policies. This means that $\pi$-PRL can interact in significantly more diverse ways (therefore less interpretable) with the environment than HC in the plots.
> > > > >
> > > > > Due to the late period within the discussion phase and compute constraints, we were only able to finish running the experiments at the moment. We present the results in Fig. 25 of Appendix D.8, and will update the potential final results by the end of the reviewing period if they finish by then.
> > > > > We note that in these results, besides the first ~200k timesteps, the SAC version of $\pi$-PRL seems to perform below HC. The parameterization that we chose for HC is able to quickly improve the quality of the policy, comparatively to the other algorithms. While it would be interesting to study the training dynamics if we let the training last for longer, we recall that even besides this control experiment, as mentioned before, the same architecture of HC is able to handle high level planning tasks, while $\pi$-PRL does not. In addition, the end-user has the possibility with HC to change its performance interpretability trade-off by increasing the number of sub-policies to use for a marginal cost by simply increasing the size of the Gumbel layer, while the same operation comes at a great computational cost for $\pi$-PRL (Qiu and Zhu, 2022, Section 3, Complexity; Orfanos et al, Section 2., paragraph 2).
> > > > >
> > > > > In summary, we thank you again for engaging with us during this rebuttal phase, and for your input that improved our paper by bringing:
> > > > > - A new experiment on the addition of L1 penalty on the weights of small actors, akin to Orfanos et al. (2023),
> > > > > - A new experiment comparing the HC to a simple distillation baseline,
> > > > > - An improvement of the literature review by including works on programmatically interpretable RL,
> > > > > - An interesting application of the $\pi$_PRL method (with TRPO) to DM Control’s Cheetah run,
> > > > > - A new experiment comparing $\pi$_PRL extended with SAC to our HyperCombinator.
> > > > >
> > > > > We think these additions greatly improved the paper and we hope that we were able to address your main concerns, and that you will consider raising your score.
> > > > >
> > > > > Thank you,
> > > > > The authors

---

> > > > > > ### Comment · Reviewer_j8Wd · 2023-11-22
> > > > > >
> > > > > > Thank you for all the effort. I agree that the paper has improved substantially since the version I first read. I have updated the rating of paper accordingly.

---

> > > > > > > ### Author Response · Authors · 2023-11-22
> > > > > > >
> > > > > > > Thank you again for all your feedback during the discussion period, and for increasing your score. The authors

---

### Official Review · Reviewer_yR55 · 2023-10-31

**Soundness:** 3 good
**Presentation:** 3 good
**Contribution:** 3 good
**Rating:** 8
**Confidence:** 3

**Summary:**

### Problem Statement

The paper addresses the challenge of crafting interpretable policies in Deep Reinforcement Learning (DRL) to foster the development of trustworthy autonomous agents. The conventional linear policies, while interpretable, lack the expressivity to tackle complex tasks. In light of this, the authors advocate for piecewise-linear policies that aim to marry the interpretability of linear policies with enhanced performance akin to complex neural models.

### Methodology

To this effect, they introduce the "HyperCombinator" (HC), a neural architecture that embodies a piecewise-linear policy with a controlled number of linear sub-policies. Every interaction with the environment engages a specific linear sub-policy, enhancing the interpretability while maintaining competitive performance. The architecture is a preset number of sub-policies, each in the form of a set of linear coefficients, gated by a MLP with one-hot output with Gumbel-softmax reparameterization allowing end-to-end training.

### Main Contributions

The key contributions encapsulated in the paper are as follows:

- They delineate the attributes that a desirable interpretable piecewise-linear policy should exhibit.
- They design the HyperCombinator, a novel architecture that encapsulates a piecewise-linear policy with a defined number of linear sub-policies, and which is amenable to a broad spectrum of RL algorithms.
- They analyze the interpretability of HyperCombinator.
- They conduct evaluations of the model on control and navigation tasks, showcasing that the model retains a robust performance despite its curtailed expressivity.
- They leverage the interpretability of HC to develop two visualizations that elucidate the policy's reactions to inputs and unveil the temporal abstractions in task execution through tracking the sequence of sub-policies employed.

**Strengths:**

### Originality and significance

The problem of interpretable Deep Reinforcement Learning is of great practical value. Despite the simplicity of the proposed approach, it attains competitive performance while providing much more interpretability compared to conventional deep neural networks. The formulation of desired properties of an interpretable policy is also insightful.

### Writing

The paper is very well written. The organization is logical, presentation accurate and efficient. First laying out the desired properties and then evaluating the method against the properties makes the paper very easy to follow.

**Weaknesses:**

- The method has limitations that prevent it from being applied to complex problems of larger scale: The policy loses interpretable as soon as the number of linear sub-policies grows large.

**Questions:**

- In the Cheetah control setup, is the action space continuous? If so, what does the "action 0" in Figure 3 mean? Is it one dimension of the action? Then it is not proper to refer to it as "for one action of $\tilde{\pi}$".
- Why are SAC and RIS chosen as the base alogrithm for RL in the experiments?
- Are subpolicies trainable? Besides the diversifying regularizations for the MLP, would similar regularizations encouraging a diversified sub-policy help with the learning and model capacity?

---

> ### Author Response · Authors · 2023-11-15
>
> We thank the reviewer for their feedback. We are glad they enjoyed reading the paper.
> HC indeed does not escape the performance-interpretability trade-off, and increasing the number of sub-policies tends to improve performance but reduce interpretability (Fig. 4). We think one of the interesting aspects of our work is that the stakeholder can decide this trade-off for themselves, by fixing how many sub-policies they are willing to use.
>
> ### Questions
> - In Fig. 3, you are absolutely right, we are referring to the computation of the logits for the dimension 0 of the action space. We have updated the legend accordingly (in red), thanks.
> - We chose SAC because it was a strong continuous control, actor-critic baseline, with several good open-source implementations easy to bootstrap from. SAC is however not adapted to the maze experiments, mainly due to the specificities of the environments such as sparse rewards. We therefore looked for a base algorithm that would (1) be adapted to maze experiments, (2) be open-source and (3) have no competing architectural assumptions on the policy. RIS fit all these categories, which is why we chose it.
> - The sub-policies are indeed trainable. We learned each sub-policy (jointly with how to choose them) in each of our experiments. We actually tried to diversify the initialization of each sub-policy earlier in the project, but we did not find any benefits in terms of performance at that time. We therefore find the reviewer’s proposition of diversifying the sub-policies themselves (in addition to how to choose them) could be very interesting to revisit in the future, as it might help each sub-policy to specialize even more.

---

> > ### Comment · Reviewer_yR55 · 2023-11-20
> >
> > I thank the authors for their response to the questions. I would maintain my rating.

---

> > > ### Author Response · Authors · 2023-11-21
> > >
> > > Thank you for your reply. Best, the authors

---

### Official Review · Reviewer_8HLj · 2023-10-31

**Soundness:** 3 good
**Presentation:** 3 good
**Contribution:** 2 fair
**Rating:** 6
**Confidence:** 3

**Summary:**

The paper proposes to make Deep RL more interpretable by training a policy which is a piecewise linear parameterization of policies. It is supposed to solve most of the set of constraints an interpretable policy should satisfy (Section 3). The method works by using a Gumbel Softmax to select the different linear polices and then interpreting predictions based on the linear models. Some experiments show the  ability of their model to converge is more or less equal to normal methods, and arguments are made that is interpretable.

**Strengths:**

The strength of this paper is that is is proposing a reasonable solution to a very difficult problem of interpretable deep RL. I like the general idea and it makes sense, and results are reasonable enough. It should also pave a way for counterfactual reasoning as the authors suggest, and several other possible future directions.

**Weaknesses:**

There are two main weaknesses of the paper in my view.

Firstly, there appears to be clear limitations of the method performance wise, it appears to really struggle to latch onto the similar abilities of the black-box, which can be problematic as reducing performance has catastrophic effects on user trust (and then appropriate reliance etc.) Or at least, HC64 etc. is needed to get reasonable performance as opposed to HC8.

Second, the authors don't actually show the method being particularly useful for anything. It isn't always necessary to do this, but I feel here it would have helped a lot. Usually, explainability methods are used to debug, teach, regulate, calibrate reliance, or offer recourse etc., but this method isn't shown to be able to do any of those things.

### Other
* There is a misspelling in the title of parameterization.
* I wouldn't limit your Section 3 to just "counterfactual reasoning", there are many other types of contrastive explanation which your method could also generate.
* I also don't fully get (P2) here.
* p6: Please don't say "remarkable", let the readers decide themselves.
* Would be good to briefly elaborate in Fig2 why Finger spin does differently.

**Questions:**

* Sorry if I missed it, but does this work for pixel-based input data?
* Do you see this being useful for debug, teach, regulate, calibrate reliance, or offer recourse etc.,? If so, why didn't you demonstrate this.

I think overall, given the positives and negatives, I lean a bit towards acceptance, but will await the rebuttal etc. to mutate my score later. Thanks.

---

> ### Author Response · Authors · 2023-11-15
>
> We thank the reviewer for their review of our work and their suggestions. We understand that the reviewer has concerns about (1) the performance of our method and (2) the interpretability that is gained with HC. We detail our answer to these concerns below.
>
> ### Performance of HC
> We agree with the reviewer that performance is an important factor in user trust. One of HC’s main advantages is that it lets the stakeholder tune the performance-interpretability trade-off through the number of sub-policies. In certain safety-critical environments or when the stakeholder is not a domain expert, the stakeholder might be willing to sacrifice some of the performance in order to gain insights about how the policy interacts with its environment.
>
> On the other hand, a domain expert might be comfortable with more complexity, unlocking higher performance. This is illustrated in Fig. 4: HC64 never catastrophically fails, despite the high difficulty of the maze environments and its relative simplicity compared to the MLP baselines. HC8 struggles to solve the harder tasks but leads to simpler visualizations (e.g., Fig. 5). We think this is especially encouraging given the high difficulty of the mazes.
> We update our manuscript in Sec 3.2 to express this point more clearly, in blue.
>
> Finally, some applications demand first and foremost the highest level of performance. We think our approach is outside of this scope, since a performance cost has to be paid for the gains in interpretability.
>
> ### Applications of the method
> We thank the reviewer for their remark. In this work, our goal was to study the properties that could improve the interpretability of piecewise-linear policies, and to propose a parametrization that let us achieve as many of these properties as possible. Therefore, we studied and evaluated the HyperCombinator as a general-purpose architecture.
>
> The two levels of interpretability of HC relate to its transparency. The linear coefficients inform of the importance of a feature in the computation made by a subpolicy, and the sequence of chosen behaviors gives insights about the structure of the inner decision process of the agent.
> In turn, a domain expert can use this transparency to better understand the policy that they roll out (Appendix D.7 gives a low-dimensional example of such an analysis).
>
> The sub-policy sequence visualization could also be used to detect issues at run-time, especially if no video of the agent is available. In Fig. 15 (Appendix D.6), we studied the robustness of HC to perturbations, which we could model as an adverse event. The perturbations that the agent is subjected to are translated into an unusual sub-policy sequence between timesteps 100 and 200. Hence, we could imagine a system that detects a change in the subpolicy sequence, which would in turn raise an alarm that an adverse event has been detected.
>
> Overall, HC opens the gate to new analyses thanks to its transparency, which we think is a significant improvement from the status quo.
>
> ### Other remarks
> - We reformulated the “remarkable” mention in p.6 following your suggestion.
> - Contrastive explanations: we thank the reviewer for their suggestion. We agree that the architecture of HC opens up new possibilities, such as analyzing why a given sub-policy was chosen rather than another.
> - P2: we clarified the meaning of the property in p.3 following your suggestion.
> - Title: we believe that both “parameterization” and “parametrization” are valid ways to spell the word.
> - Finger spin consists in the very quick repetition of the same precise movement in order to maximize the reward. Policies with more degrees of freedom have an advantage here as they can reach a better precision thanks to their higher expressivity. We find this not too surprising, as complex policies end up performing better than HC given enough samples (Fig. 12, Appendix D.4). We added a remark about this phenomenon in Appendix D.4.
>
> ### Questions
> It is indeed entirely possible to use interpretable features while conditioning the Gumbel network on pixel data. For instance, in DM control, we could define the observations as multimodal: one pixel representation of the environment, and the associated (interpretable) proprioceptive features. The Gumbel network could be fed the pixel rendering of the environment, which would predict a sub-policy to apply. This sub-policy would be linear with respect to the interpretable features. We did not propose results with pixel data for now as we felt this was beyond the scope of the paper, which introduces the architecture with respect to interpretable features only. However, we do agree it would be an interesting follow-up, and intend on pursuing a pixel-based version of HC in future work.

---

> ### Author Response · Authors · 2023-11-20
>
> Dear reviewer,
>
> Thank you again for the time you already dedicated to reviewing our work.
>
> As the end of the discussion period approaches, we are reaching out again to check if our previous reply answered your concerns. If not, please let us know, as we would be happy to have the opportunity to address your remaining concerns.
>
> The authors

---

> ### Comment · Reviewer_8HLj · 2023-11-20
>
> Hi, thanks a lot for your reply, I have read everything and thought it over for a while.
>
> I think I can agree with most of what you say, the only part I feel we part ways is about applications of the technique. I would like to offer potentially useful ideas for you.
>
> You say *"In turn, a domain expert can use this transparency to better understand the policy that they roll out (Appendix D.7 gives a low-dimensional example of such an analysis)."*
>
> This is a good start, but why exactly do you want to understand the policy? That's the key question. If it is for debugging purposes, then you need a user study showing it does this. If it is to calibrate appropriate reliance, then again you need users in the loop to show it actually does this, otherwise it's all conjecture.
>
> *The sub-policy sequence visualization could also be used to detect issues at run-time, especially if no video of the agent is available*
>
> Again, I think you need to show how seeing these issues is helpful. I know it sounds obvious that it should be useful, but too often XAI techniques are published with the author's claiming the method is "interpretable", but all the while failing to demonstrate that the method is understandable and useful to intended practitioners of the system.
>
> I like the paper, which is why I accepted it, but I cannot raise my score any higher with this fundamental issue present. Thanks again and good luck!

---

> > ### Author Response · Authors · 2023-11-21
> >
> > Thank you for your reply and for the additional context about your decision. The authors

---

### Official Review · Reviewer_KsbR · 2023-11-06

**Soundness:** 2 fair
**Presentation:** 3 good
**Contribution:** 2 fair
**Rating:** 3
**Confidence:** 3

**Summary:**

The authors propose a framework for interpretable reinforcement learning that attempts to balance interpretability, performance, and computational complexity. They propose learning a set of piecewise linear policies, which have the benefit of interpretability of linear models. They ensure that the number of piecewise linear policies is not too large which would inhibit practical interpretability. They present the empirical performance of their framework, HyperCombinator, in control and navigation tasks.

**Strengths:**

- The communication in the paper is clear. The authors clearly describe the desiderata for the ideal interpretable RL methods, and are clear about how their proposed approach seeks to address these points.
- The authors' commitment to maintaining linear policies at some level is good, as these will always be very interpretable on their own.
- The authors provide nice visualizations of their approach and their empirical results.

**Weaknesses:**

- The contribution level is low in this paper. Pasting together a set of linear policies does not seem that differentiated from prior work.
- It's not clear practically how interpretable the resulting model is. I suppose the user is supposed to inspect the linear model coefficients to understand what the policy is doing. However, there is not much discussion of this. For example, how should I interpret the coefficient heatmap in Figure 3?
- As it stands, it seems the approach is less of a compromise between interpretability and performance and more of a deterioration of performance while only gaining a little bit of interpretability.

**Questions:**

How complex of environments can the HC framework handle without fully losing performance?

---

> ### Author Response · Authors · 2023-11-15
>
> We thank the reviewer for reviewing our submission and for their feedback. We understand that the reviewer has concerns about (1) the novelty of our work and (2) the resulting interpretability and the trade-off with performance. We address these concerns below.
>
> ### Novelty of our approach
> There are several components that distinguish our paper from previous works. Notably, several aspects of the Gumbel network, the module which decides which sub-policy to apply to a given input, are key to let HC learn in the different experiments:
> - In particular, the MLP in the Gumbel network leads to an intricate partition of the input space (unlike regular tree methods), which enables a good performance of HC in the DeepMind Control benchmark as well as in select mazes.
> - Moreover, we specifically use the straight-through estimator (STE) to guarantee the selection of a single sub-policy at each timestep (instead of a mixture of several sub-policies), during training and during evaluation. This is an important property for interpretability (enabling analyses during training) that was missing from previous works.
> - Finally, while previous works (such as Akrour 2018) focused mostly on performance, we study in detail how such a parameterization influenced the piecewise-linear functions we could express, within the scope of interpretability. The different visualizations give insights about how the agent is interacting with its environment, which are not possible with regular neural networks.
>
> To summarize, our paper performs an in-depth study of the limits of MLP with ReLU activations as an interpretable parametrization of piecewise-linear policies. We propose an architecture that parametrizes a piecewise-linear policy along the constraints. We think this fills a gap that was present in the literature.
>
> ### Characterization of the interpretability gained with HC
> The HC architecture provides two levels of interpretability:
> 1. Low-level interpretability through the value of the linear coefficients of each sub-policy,
> 2. High-level interpretability by examining the sequence of choices of sub-policies over a trajectory.
>
> The heatmap in Fig. 3 is an example of low-level interpretability. It shows the coefficients of each sub-policy (one column per sub-policy) for one dimension of the action space. For instance, subpolicy #0 assigns a positive coefficient to feature f0 as the corresponding cell is red. Therefore, we know that inputs with greater positive values for f0 will lead to greater predicted actions, all else being equal. Moreover, the coefficient quantifies the exact importance of feature f0 in the computation of the action.
>
> In addition, we perform a more extensive analysis of the meaning of the coefficients in Appendix D.7, where we analyze the learnt policies on the cartpole environment. The lower dimensionality of the environment helps getting a better grasp of the computations and therefore of each sub-policy.
> For instance, the coefficient values let us understand the usage of each sub-policy and how the agent specialized some to get the CartPole up, and some to maintain it straight.
> We think this is a considerable improvement over the regular non-interpretable neural network, and makes the decision making process significantly more transparent.
>
> The high-level interpretability brought by our agent is complementary to this low-level interpretability, as we get a better understanding of how the agent processes the information to decide which subpolicy to follow. For instance, Fig. 3 (left) illustrates how the agent learnt to chain efficiently the linear sub-policies in a repeated manner to solve Half-Cheetah with a strong performance, and the hierarchical structure in Fig. 5 reflects the structure of the maze.
>
> ### Question about interpretability-performance trade-off
> While the performance of HC64 decreased with the increasing difficulty of the mazes, it never collapsed (Fig. 4). We think this is especially encouraging as these tasks are extremely challenging, and require dedicated algorithms to solve them (Chane-Sane 2021). This leads us to think that HC can handle complex environments without fully losing performance simply by increasing the number of sub-policies available. For instance, in Fig. 4, we had to trade-off some interpretability by going from 8 sub-policies to 64.
>
> Therefore, the answer to this question ultimately lies with the stakeholder that needs to understand the sub-policy. If that person is a domain expert, it is possible for them to increase the number of sub-policies and therefore increase the performance.
> A person less familiar with the environment might want to reduce the number of sub-policies that HC can use, sacrificing some of the performance to understand how the agent interacts with the environment.
> We updated the manuscript (Sec 3.2) to reflect this point (in olive).

---

> > ### Author Response · Authors · 2023-11-15
> > **References**
> >
> > Akrour 2018, Regularizing reinforcement learning with state abstraction, International Conference on Intelligent Robots and Systems (IROS), pages 534–539. IEEE, 2018
> >
> > Chane-Sane 2021, Goal-conditioned reinforcement learning with imagined subgoals. International Conference on Machine Learning, pages 1430–1440. PMLR, 2021.

---

> ### Author Response · Authors · 2023-11-20
>
> Dear reviewer,
>
> Thank you again for the time you already dedicated to reviewing our work.
>
> As the end of the discussion period approaches, we are reaching out again to check if our previous reply answered your concerns. If not, please let us know, as we would be happy to have the opportunity to address your remaining concerns.
>
> The authors

---

### Meta-Review · Area_Chair_TBZU · 2023-12-09

**Metareview:**

This paper proposes a novel approach to interpretable reinforcement learning that uses a piecewise-linear neural architecture to express the policy in terms of a small number of interpretable sub-policies. The authors propose a novel architecture, the HyperCombinator, that can learn a relatively complex partition of the input space into different sub-policies while learning independent parameters for each sub-policy (in contrast, an MLP "mixes" the parameters for the input space partition with those of the sub-policies). The authors demonstrate that their approach achieves good performance in their experimental evaluation.

The reviewers generally agree that the problem being studied is interesting an important, and that the authors have a compelling experimental evaluation. In particular, the authors help address an important question in interpretability of how to decompose complex tasks into simpler ones, and introduce a new architecture based on Gumbel-Softmax for doing so. While there were some missing baselines, especially from the programmatic reinforcement literature, the authors have added experiments along these lines. There are also concerns about the definition and meaning of interpretability; the authors might be able to more convincingly demonstrate interpretability by demonstrating the ability to diagnose issues in underlying policies. However, this remains a challenge for the interpretability literature, and is not specific to this work. Finally, there were some concerns about the novelty of the approach.

**Justification For Why Not Higher Score:**

While the authors make an interesting contribution, their approach has limited novelty and they leave open questions about whether their approach is truly interpretable.

**Justification For Why Not Lower Score:**

The authors solve an important problem using an interesting approach, and demonstrate that their approach achieves good performance in their experiments.

---

### Decision · Program_Chairs · 2024-01-16

Accept (poster)